# Predatory and Defensive Strategies in Cone Snails

**DOI:** 10.3390/toxins16020094

**Published:** 2024-02-07

**Authors:** Zahrmina Ratibou, Nicolas Inguimbert, Sébastien Dutertre

**Affiliations:** 1CRIOBE, UAR CNRS-EPHE-UPVD 3278, University of Perpignan Via Domitia, 58 Avenue Paul Alduy, 66860 Perpignan, France; zahrmina.ratibou@univ-perp.fr; 2IBMM, University of Montpellier, CNRS, ENSCM, 34093 Montpellier, France

**Keywords:** *Conus* species, conotoxins, “milked” venom, predatory and defensive venom, motor, nirvana, lightening-strike cabals

## Abstract

Cone snails are carnivorous marine animals that prey on fish (piscivorous), worms (vermivorous), or other mollusks (molluscivorous). They produce a complex venom mostly made of disulfide-rich conotoxins and conopeptides in a compartmentalized venom gland. The pharmacology of cone snail venom has been increasingly investigated over more than half a century. The rising interest in cone snails was initiated by the surprising high human lethality rate caused by the defensive stings of some species. Although a vast amount of information has been uncovered on their venom composition, pharmacological targets, and mode of action of conotoxins, the venom–ecology relationships are still poorly understood for many lineages. This is especially important given the relatively recent discovery that some species can use different venoms to achieve rapid prey capture and efficient deterrence of aggressors. Indeed, via an unknown mechanism, only a selected subset of conotoxins is injected depending on the intended purpose. Some of these remarkable venom variations have been characterized, often using a combination of mass spectrometry and transcriptomic methods. In this review, we present the current knowledge on such specific predatory and defensive venoms gathered from sixteen different cone snail species that belong to eight subgenera: *Pionoconus*, *Chelyconus*, *Gastridium*, *Cylinder*, *Conus*, *Stephanoconus*, *Rhizoconus*, and *Vituliconus*. Further studies are needed to help close the gap in our understanding of the evolved ecological roles of many cone snail venom peptides.

## 1. Introduction

Cone snails are specialized carnivorous marine mollusks that can be found in coral reef areas, from shallow intertidal to deeper waters, and spread across the tropical Indian, Pacific, and Atlantic Oceans [1]. They are classified as gastropods within the Conidae family, which feature hollow radular teeth and venom glands [2]. They use a complex venom mixture to paralyze and hunt fish, mollusks, and worms [3]. This venom is secreted through epithelial cells lining the cone’s venom gland, which is a long and thin tubular duct [4]. A singular radular tooth, analogous to a hypodermic needle, is then moved into the proboscis through which the rapid-acting venom is injected. The venom is acknowledged as a rich source of potent pharmacological components, raising high interest in the drug development field [5].

This venom consists primarily of biologically active peptides, generally characterized as conotoxins or conopeptides. They can be classified into two groups: conotoxins, which are cysteine-rich conopeptides consisting of 10 to 30 amino acids, while conopeptides are cysteine-poor, meaning 1 or no disulfide bond [4,5,6]. Moreover, conotoxins are highly structured and often show high affinity and selectivity toward membrane receptors, ion channels, and other transmembrane proteins of the nervous and non-nervous systems [4]. Conopeptides include several types of cysteine-poor peptides, such as contulakins, conantokins, conorfamides, conolysins, conophans, conomarphins, contryphans, conopressins, and more recently, hormone-like conopeptides, such as elevenins or prohormones [5,7]. Conopeptides are usually minor in comparison to conotoxins in the venom mixture and each presents a selective type of target [7]. These small peptides can work as ligands, which induce a physiological reaction by interacting with a given receptor [4]. Conotoxins and conopeptides are secreted as peptide precursors, which can be portioned into three characteristic sections: a highly conserved signal peptide, representative of the gene superfamily from which it was translated, a pro-peptide section, and a highly diversified mature peptide (Figure 1). The mature peptide is the active sequence portion, which is enzymatically cleaved and then modified into a highly stable structure within the injected venom [8].

The cysteine pattern within the conotoxin sequence is designated with roman numerals and it directs the tridimensional structure, which in turn also influences their biological activity. So far, although only few conotoxins have been fully characterized pharmacologically, more than 20 pharmacological targets have been identified. Some of the biological targets involve, for the most part, ion channels, but also some G-protein-coupled receptors and transporters [3]. Conotoxins are classified according to their targets into pharmacological families, defined by Greek letters, such as α, δ, μ, ω, κ, γ, etc. (Figure 2) [3]. For instance, ω-conotoxins are antagonists of voltage-gated calcium channels, and some are effective against neuropathic pain [3]. Such activity was the basis for the development of the first marine-based drug isolated from a cone snail, known as Prialt^®^. This drug is a synthetic version of the ω-conotoxin MVIIA isolated from the piscivorous species, *Pionoconus magus* [9]. Likewise, some α-conotoxins have been characterized as nicotinic acetylcholine receptors (nAChRs) antagonists, with some of them having potential in the treatment of pain, cognitive, cardiovascular, and other disorders [9]. For the past three decades, research in the field has been mainly focused on finding new ligands for known targets, with a strong emphasis on modulators of pain receptors [9].

The ~800 species of cone snails can be categorized into three main groups according to their diet. Piscivorous species hunt fish, molluscivorous species prey upon mollusks, and vermivorous species feed upon worms (Figure 3). The type of radula tooth seems to be directly correlated to the diet, and this criterion has been used to support the classification of species [11]. Based on molecular phylogenetic studies, cone snails have been classified into a single large family, Conidae, which can then be divided into four genera: *Conus*, *Conasprella*, *Profundiconus*, and *Californiconus* [2]. The genus *Conus* constitutes more than 85% of all cone snail species, which can then be further classified into 57 subgenera or ‘clades’ of *Conus* species, which represent a clear subgrouping within the genera [2]. These classifications can provide a better understanding of the “biotic interactions” within *Conus* species [4]. Unfortunately, rather than being tested on biologically relevant animal models, cone snail venoms have almost exclusively been investigated using mammalian bioassays. As a result, the conclusions drawn from these assays should be interpreted with caution when extrapolated to the biology of cone snails.

### 1.1. Envenomation Strategies in Cone Snails

Piscivorous cone snails exhibit varying types of hunting behaviors. For instance, upon the detection of a prey, first through chemosensory cues [12], some cone snails extend their proboscis in order to inject a paralytic venom (Figure 4A). The venom is injected via a radula tooth that is comparable to a miniature harpoon that the cone snail uses to sting and tether the prey to avoid its escape [4,13]. Upon the strike, the prey often displays an immediate tetanic paralysis. The cone snail then retracts its proboscis to drag its victim toward its enlarged rostrum to engulf it [13]. The archetype of this behavior is the ‘taser-and-tether’ strategy employed by the majority of piscivorous species from the *Pionoconus*, *Textilia*, and *Chelyconus* clades, where injection of venom first produces an immediate paralysis (“taser”), followed by the reeling back of the tethered fish into the rostrum via the contraction of the proboscis, which is still tightly grasping the base of the radula tooth [4].

Remarkably, some other cone snails have been observed to catch their prey without prior sting. In this case, the cone snail is hypothesized to release a set of toxins in the water, which places the prey into a sedative-sleepy state (Figure 4B). The cone snail then opens its rostrum to engulf it and may proceed to envenomate and predigest the prey [13]. Thus, cone snails that use this strategy, named as ‘net-hunting’, would supposedly release venom components in the water and inject paralytic peptides, which induces an irreversible neuromuscular paralysis of the captured prey. Lastly, the “strike-and-stalk” envenomation strategy is a variation of the taser-and-tether strategy, where the cone snail strikes a prey without tethering it and engulfs it after immobilization has occurred. The latter strategy remains less studied in terms of the neurobiological mechanism involved [13].

In non-piscivorous cone snails, the hunting behaviors have been much less investigated. For most molluscivorous species observed in captivity or in the wild, the predatory strategy involves actively chasing the prey and injecting, multiple times, fine, arrow-like radula teeth into the foot of the prey [14]. The firing of the radula tooth is usually accompanied by vigorous pumping of copious amount of venom, which can be seen, when injected in excess, as a whitish cloud escaping out the tip of the proboscis and/or out of the base of the tooth from back pressure [15]. In the case of the mass spectrometry (MS) analysis of successive stings by *Cylinder textile*, modest variations in the venom composition were described [16]. The first injection usually stops or slows down the prey but does not completely incapacitate it; therefore, it was suggested that a second, third, or more injections, possibly with different peptides, were needed to eventually overcome the prey.

Hunting behaviors for vermivorous species are even more elusive, except for only a few species. Both *Stephanoconus imperialis* and *Stephanoconus regius* prey almost exclusively on amphinomid worms (“fireworms”). These two species use a prey capture strategy reminiscent of the “taser-and-tether” strategy employed by many piscivorous species. Indeed, the targeted worm is first detected by the chemosensory organs, inducing the extension of a reddish proboscis. The short radula tooth (1–1.5 mm) is then fired and embedded into the worm’s body, forcefully pushing through a remarkable quantity of a greening venom (Figure 3C) [9]. As described for the fish-hunters, the envenomated prey shows immediate involuntary contractions, leading to incapacitation, and is reeled back into the rostrum. Our personal observations on other vermivorous species often reveal, surprisingly, an apparent venom-less strategy, where the snail directly attempts to swallow the worm through its extended rostrum without prior stinging via the proboscis. One of the most mysterious prey strategies relates to the vermivorous species hunting tube worms, as there is no description in the literature.

### 1.2. Reality Check on the Concept of Cabals

Early pharmacological characterization of conotoxins from venom gland extracts revealed a variety of targets and modes of action. From the pharmacological effects obtained mostly on mammals, extrapolations were made to explain the effects observed on prey, and this is how the concept of cabals was first crafted. The cabals are defined as a group of (artificially put together) conotoxins, which seem to modulate the same physiological target or may act synergistically. Thus, the “lightening-strike cabal” is defined as a set of κ- and δ-conotoxins, as well as conkunitzins, which would together elicit an excitatory state on the prey [17,18]. This reaction is due, respectively, to the inhibition of K^+^ channels, as well as a delayed inactivation of Na^+^ channels [4].

Meanwhile, the “nirvana cabal” is highly speculative, but could include the release in the surrounding water of a mixture of B1-conotoxins [19] and hormone-like peptides that would induce a “hypoactivity in sensory neuronal circuity” [4,20,21]. Although prey capture observations of net-hunting species seem to corroborate this hypothesis, there is currently no direct evidence to support any release of venom into the water. Lastly, an additional “motor cabal” was proposed to be responsible for the final flaccid paralysis that prevents the prey from recovering the initial excitatory shock. The latter involves α-, µ-, and ω-conotoxins that interfere with the neuromuscular junction [18].

Although these cabals were logically formulated, do they actually correspond to the reality of the predatory strategies employed by cone snails to defeat their prey? Nearly thirty years ago, an ingenious procedure, now commonly referred to as “milking”, was devised that allows for the collection of the injected venom, providing a direct means of interrogating the conotoxin cocktail used for prey capture [22]. Using a live prey to arouse the cone snail and trigger a predatory behavior, a microcentrifuge tube covered with parafilm, and a piece of the prey’s tissue, is presented to the tip of the extended proboscis. Sensory cilia at the tip of the proboscis identify the tissue as “prey” and instantaneously trigger the injection of venom through the radula tooth. Such recovered “milked venoms” can now be analyzed, and the composition revealed. Over the last two decades, the more milked venoms were investigated, the less obvious the role of the conotoxins described in these cabals was for prey capture [23].

Overall, in all cases investigated, milked venoms appear significantly less complex compared to dissected gland extracts. For instance, in *Pionoconus* species, the predatory venom is usually dominated by one class of conotoxins (sometimes the only conotoxins seemingly injected), the κA-conotoxins [24,25,26]. Therefore, it appears that κA-conotoxins are responsible for the immediate “taser” effect in this clade, not a combination of κ- and δ-conotoxins, as originally described for the lightning-strike cabal. Indeed, injection of κA-conotoxins alone into fish recapitulates the tetanic paralysis observed during prey capture [27]. However, it has to be noted that intraspecific variations in the injected venom can be dramatic and, occasionally, the paralytic peptides from the “motor cabal” are detected, suggesting that they could play a significant role in prey capture [23]. Although not fully explained at the time, one aspect of this diversification was later attributed, at least in part, to the unsuspected ability of some cone snails to produce two types of venoms [28].

### 1.3. Defensive Strategies

From the three dozen human deaths reported, it has long been known that cone snails can also inject their venom defensively [29]. In the literature, there is only anecdotal information on the natural predators of cone snails, but fish, mollusks (octopi), and some crustaceans are known to prey on them (Figure 5). For instance, a rare species of deep-water cone snail was first described only from a shell recovered from the stomach content of a large fish (personal communication). The defensive use of venom provides an obvious evolutionary advantage. Indeed, avoiding being eaten is one of the most important fitness-related criterion for the survival of a species, together with being able to feed and reproduce. In fact, some venomous animals only use their venom defensively (some hymenopterans, fish, etc.), whereas the reverse is not true, suggesting that the defensive use of venom may actually have a stronger evolutionary role than anticipated, possibly more than predation in some cases [30].

Thanks to their capacity to defend themselves, some species of cone snails have evolved some unique behaviors. However, for most species, the first line of defense is usually to retract deeply into the shell, which offers a strong and often inviolable fortress (Figure 6C). Others will respond aggressively to any threat by extending their proboscis (Figure 6D). If the threat intensifies, the cone snail will inject venom into the aggressor, but there are also reports of cone snails squirting venom (personal observations). Additional behavioral studies are needed to fully decipher the complex defensive responses displayed by cone snails.

The most dangerous species to humans, *Gastridium geographus*, displays an unusually aggressive behavior, and will readily use its venom defensively when handled. There seems to be a striking relationship between the fragility of the shell (as in the case of *Gastridium geographus*) and the propensity to use venom defensively. Typically, large vermivorous species will often be unfazed by any threat, being protected by heavily built shells and narrow apertures [28]. However, many species were reported to inflict injuries to humans, regardless of their diet, with varied degrees of consequences. From the known human *Conus* envenomation, various levels of severity were distinguished, from fatal to minor effects, comparable to bee stings, and the most adverse symptoms were attributed to piscivorous cone snails, especially *Gastridium geographus* [29].

The first investigation of a defense-evoked venom uncovered an unsuspected twist in cone snail biology [28]. Indeed, the defensive venom of *Gastridium geographus* was highly complex and contained massive amounts of paralytic conotoxins from the “motor cabal”, explaining the lethal symptoms in humans, as opposed to the predatory venom, which was devoid of these and instead contained prey-specific conotoxins with no activity on human receptors. Therefore, in this iconic species, paralytic conotoxins directed to the neuromuscular junction are essentially defensive weapons, not part of the prey capture strategy, a result in conflict with the cabal narrative. From this initial discovery, more data on different species were needed to evaluate how widespread this separate evolution of predatory and defensive venoms is among cone snail species. Triggering and collecting defensive venom can be achieved through different means, including using a natural predator (i.e., a molluscivorous species, such as *Conus marmoreus* or *Cylinder textile*), applying pressure to the shell, or pinching the foot of the cone (Figure 6D) [28].

Overall, the remarkable ability of cone snails to purposefully modify their venom composition upon different triggering stimuli (predatory or defensive) offers novel and unprecedented research opportunities. Indeed, separately collecting each venom type will allow unambiguous interpretation of the ecological and evolutionary roles of each conotoxin. In this review, we will describe the reported predatory- and defense-evoked venoms of 16 species belonging to eight clades of the *Conus* genus, with three being piscivorous (*Pionoconus*, *Chelyconus*, and *Gastridium*), two molluscivorous (*Cylinder* and *Conus*), and three vermivorous (*Stephanoconus*, *Rhizoconus*, and *Vituliconus*), and discuss the work that remains in order to better understand venom–ecology relationships in cone snails.

## 2. Piscivorous Cone Snails

### 2.1. Predatory Venom

Thus far, the *Conus* genus counts around 800 different species, representing about 70% of vermivorous, 20% piscivorous, and 10% of molluscivorous cone snails [31]. Over a hundred piscivorous cone snails have been classified into the following clades: *Afonsoconus*, *Asprella*, *Chelyconus*, *Embrikena*, *Gastridium*, *Phasmoconus*, *Pionoconus*, and *Textilia*, although the piscivorous diet requires confirmation for the *Afonsoconus*, *Asprella*, and *Embrikena* clades [13]. In comparison, the venoms of fish-hunting cone snails, such as *Pionoconus striatus*, *Gastridium geographus*, and *Chelyconus purpurascens*, have been extensively characterized against the prevailing vermivorous species (Figure 7) [28,32,33].

In general, the predatory venoms of fish-hunting cone snails show major contributions of small disulfide-rich conotoxins over larger ones and cysteine-poor conopeptides, such as conophysins [28], conopressins [28,33], and contryphans [32,35] (Figure 8A). The conotoxins identified are scattered into a dozen gene superfamilies, dominated by A-, O1- and M-conotoxins. The rest are attributed to the I3, C, O2, T, and B1 superfamilies [23,29,33]. Most conotoxins that were identified in the predatory venoms of fish-hunting cone snails were previously biologically characterized from venom gland extracts. The α-, κA-, δ-, κ-, μ-, and ω-conotoxins constitute the major pharmacological families identified in the predatory venom (Figure 8B). κA-conotoxins are the most abundant (relative contribution to the injected venom) (Table 1), but α-conotoxins are the most prevalent (in terms of number of sequences identified) in the predatory venoms of fish-hunting cone snails (Table 2). These conotoxins are especially represented in the *Pionoconus* and *Chelyconus* clades, and less in the *Gastridium* clade.

As mentioned, the superfamily A is the most represented in the predatory venom thanks to κA-conotoxins and α-conotoxins. κA-conotoxins were first discovered in the predatory venom of *Pionoconus striatus*, with κA-SIVA and κA-SIVB [32], and their short and non-glycosylated equivalent κA-PIVE and κA-PIVF were identified in *Chelyconus purpurascens* (Table 1) [17,38]. Exhaustive investigation of the predatory venom of *Pionoconus consors* also revealed the importance of κA-conotoxins, with the abundant injection of κA-CcTx and the sequencing of a series of CcTx variants [24,37,50]. Although not confirmed, the major compounds found in the predatory venom of another *Pionoconus* species, *Pionoconus magus*, were determined within the mass range of κA-conotoxins [40]. More recently, these κA-conotoxins were identified abundantly in both predatory and defensive venom of *Pionoconus striatus* [32] and in the predatory venom of *Pionoconus catus* [25]. Interestingly, a recent study has identified a variant of the glycosylated κA-conotoxins (κA-SIVC) in the predatory venom of specimens of *Pionoconus striatus* from Mayotte (France), suggesting that geographical variations can be population-specific [36]. κA-conotoxins were initially characterized as excitatory peptides that block K^+^ channels, yet controversy remains over the molecular target since Na^+^ channels were also suggested as the targeted receptor [27]. Although they uphold the same IV cysteine framework as certain αA-conotoxins, such as α-OIVA (Table 2), their activities are different: the firsts are excitatory while the seconds are not [41]. Generally, these κA-conotoxins appear as the major and most abundant component in the predatory venom of *Pionoconus* species and are likely solely responsible for the rapid immobilization of prey.

Next, α-conotoxins (targets are the nAChRs) are the most prevalent pharmacological family in terms of number of sequences identified in the predatory venoms of fish-hunting cone snails. These α-conotoxins, although not systematically injected, are especially represented in the *Pionoconus* and *Chelyconus* clades, and less in the *Gastridium* clades [2]. For instance, some of the α-conotoxins identified in the *Pionoconus* clade include α-SI (*Pionoconus striatus*) [23,32,39], α-CnIB (*Pionoconus consors*) [37,51], α-MI (*Pionoconus magus*) [40], and α-CIB (*Pionoconus catus*) [25]. Similarly, α-PIB (*Chelyconus purpurascens*) and α-EIIA (*Chelyconus ermineus*) [52] are found in the *Chelyconus* clade. Finally, the *Gastridium* subgenus shows the least amount of identified α-conotoxins in their predatory venom, with only α-GIC (*Gastridium geographus*) [33] and α-OIVA (*Gastridium obscurus*) confirmed so far [41]. When injected into a prey, some of these α-conotoxins operate as slow (several minutes) paralytics analogous to the snake α-neurotoxins, as they selectively target and inhibit the muscle type of nAChRs (Figure 2) [10,53]. Interestingly, the subtle variations in length and amino acid residues determine the targeted site or nAChR subtype [10]. Usually, small 3/5 (3 and 5 correspond to the number of residues in inter-cysteine loops) α-conotoxins, such as α-SI, α-MI, and α-CnIB, selectively target muscle-type nAChRs [5], whereas larger 4/7 α-conotoxins, such as α-MII and α-GIC, target neuronal nAChRs, and their role in prey capture is less understood [43]. Considering their potentially useful paralytic capacities, and according to the “motor cabal” hypothesis, their presence in the predatory venoms would seem compulsory, but analysis of individual predatory venom rather than pool of collected venom suggests otherwise [10].

Exceptionally, δ-conotoxins were also detected in the predatory venom of some fish-hunters of the *Pionoconus* and *Chelyconus* clades (Table 3). This family of conotoxins was characterized as voltage-gated sodium channel (VGSC) modulators. Indeed, δ-conotoxins activate Na_v_ channels via a delay in the inactivation mechanism (Figure 2) [54]. δ-Conotoxins, like the κ-conotoxins, are defined as excitatory peptides, which induce the rapid tetanic paralysis in their prey [32]. For example, the δ-PVIA isolated from *Chelyconus purpurascens* was characterized as the “lock-jaw peptide” because it causes a rigid paralysis of the prey, particularly visible around the mouth musculature [55]. Again, considering their critical role in the “lightning-strike cabal”, these conotoxins are expected to be always injected for prey capture, but it is almost never the case.

Very few μ-conotoxins have been found in the predatory venom of three piscivorous clades: *Pionoconus consors*, *Gastridium geographus*, and *Chelyconus purpurascens* (Table 4). The molecular targets of μ-conotoxins are VGSCs, more precisely Na_v_ channels, which play an important role in the central and peripheral nervous system (CNS and PNS) (Figure 2). However, contrary to δ-conotoxins, they act as blockers of the channel, instead of delaying its inactivation. The μ-conotoxin-GS is the only μ-conotoxin identified from *Gastridium geographus* predatory venom, and is highly potent on fish, but less on mammalian Nav channels [57]. Another example is the μ-CnIIIC, isolated from the venom of *Pionoconus consors*, from which a synthetic version was commercialized as a cosmetic to smoothen facial lines (XEP^TM^-018) [58]. The μ-CnIIIB was shown more specifically to block tetrodotoxin-resistant (TTX-R) Na channels, where the blockade was observed as “slow and reversible” [59]. Another type of μ-conotoxins, the μO-Conotoxins, restrict channel opening, which has been associated with many side effects upon intravenous application, such as paralysis and death of models in inflammatory and neuropathic pain, leading to a discouragement of further research as therapeutic agents [10]. 

Trace amounts of ω-conotoxins known as “shaker” peptides have been detected in the predatory venoms of some fish-hunters [60]. ω-Conotoxins belong to the O1 gene superfamily with a VI/VII framework (Table 5). They are able to selectively block N-type Ca^2+^ channels (at the nerve terminals) and prevent the release of important neurotransmitters (such as glutamate, GABA, acetylcholine, dopamine, etc.) [61]. They are pore blockers, which means that they can physically block the influx of Ca^2+^ ions [62]. Upon intracerebral injection, they produce persistent shaking in mice [60]. ω-Conotoxins were extensively investigated for their potential in pain treatments, including ω-MVIIA, which is the only FDA-approved conotoxin drug (from *Pionoconus magus*, known as Ziconotide), as well as ω-CVID (Leconotide) isolated from *Pionoconus catus*, which is also under development for pain treatment [63]. Few ω-conotoxins have been detected in the predatory venom of *Pionoconus* cone snails, such as *Pionoconus striatus* (ω-SVIA and ω-SVIB) [39,64], *Pionoconus consors* (ω-CnVIIA) [37,51], and *Pionoconus catus* (ω-CVIA) [25,61].

Finally, some conopeptides, larger conotoxins, and proteins were also isolated from predatory venoms of fish-hunting cone snails (Table 6). These include contryphans, detected in *Gastridium geographus*, *Chelyconus purpurascens*, and *Pionoconus striatus*. Contryphans are known as Ca^2+^ channel modulators. For instance, contryphan-P (*Chelyconus purpurascens*) was associated with the “stiff-tail” syndrome when injected in mice [54]. Contryphan-S was found in the venom of *Pionoconus striatus* but was annotated as contryphan-G, as they present identical sequences [32]. Both contryphan-S and -G seem to be used for preying purposes by *Pionoconus striatus* and *Gastridium geographus* [28]. Moreover, *Gastridium geographus* injects non-paralytic compounds, including conopressin, conophysin, contulakin, and conantokin [28,33]. Conopressins and conophysins have been shown to have agonist/antagonist activity against vasopressin receptors [65], while conantokin-T was deemed responsible for the sleep-like state of the prey prior to being engulfed, especially for the “net-hunting” cone snails (Figure 4), which is caused by the inhibition of NMDA (N-methyl-D-aspartate) receptors [54,66]. Contulakin-G, on the other hand, is a glycopeptide identified as an agonist of neurotensin receptors, which has shown analgesic properties [67]. Interestingly, *Pionoconus striatus* also injects larger peptides, such as the conkunitzins, which have been characterized as voltage-gated K^+^ channel blockers, affiliated with the Shaker potassium channels [68]. Lastly, large polypeptides, such as p21a (*Chelyconus purpurascens*) [69], or con-ikot-ikot (*Pionoconus striatus*) [70] that inhibits AMPA (α-amino-3-hydroxy-5-methyl-4-isoxazole propionic acid) receptors, as well as hyaluronidases (*Pionoconus consors* and *Chelyconus purpurascens*) [37,71], phospholipases A2 (*Chelyconus purpurascens*) [72], and proteases (*Chelyconus purpurascens* and *Chelyconus ermineus*) [73], have been identified in predatory venoms. More high-molecular-weight proteins, such as metalloproteases, are likely to be present, but their role in prey capture remains unclear [74].

### 2.2. Defensive Venom

The data on defensive venom are more scarce compared to predatory venoms. Indeed, so far only *Pionoconus striatus* and *Gastridium geographus*, as well as *Gastridium obscurus*, have been investigated regarding their defense strategies [32,33]. Six gene superfamilies and three conopeptide classes (Figure 9A) have been identified in the defense venoms. Most of them are A-, O1-, and M-conotoxins found in both *Pionoconus* and *Gastridium* clades. In addition, B1-, S-, and T-conotoxins are found exclusively in *Gastridium* cone snails, as well as conophysins. Moreover, *Pionoconus striatus* also injected conkunitzin and con-ikot-ikot (Table 6).

Overall, five pharmacological families were characterized from the defensive venom of these cone snails: α, ω, μ, κ, and σ (Figure 9B). The α-conotoxins α-SI and α-SIA from *Pionoconus striatus* and α-OIVA and α-OIVB from *Gastridium obscurus* are used, both in the predatory and defense venoms. However, *Gastridium geographus* injects a specific set of α-conotoxins, including α-GIA, α-GII, and α-GID, all with a I cysteine framework [33]. Targeting neuronal rather than muscle-type nAChRs, α-GID has an unusual 4-residue N-terminal tail, which appears critical for activity at the α4β2 nAChR subtype but not the others (α3β2 or α7) [75]. In the defense venom of *Pionoconus* species, κA-conotoxins, such as κA-SIVA and κA-SIVB in *Pionoconus striatus*, are found abundantly [32].

In contrast to the predatory venom, we observe that *Gastridium geographus* uses large amounts of μ-conotoxins μ-GIIIA, μ-GIIIB, and μ-GIIIC [28,33] for defense purposes, but not the prey-specific μ-conotoxin-GS (Table 4). Moreover, *Pionoconus striatus*, which lacked μ-conotoxin in the predatory strategy, injects one in the defensive venom (μ-S3-G02) [32]. Defense venoms also include ω-conotoxins (Table 5). In addition to ω-SVIA and ω-SVIB, *Pionoconus striatus* injects two O1-conotoxins, SO4 and SO5, in rather high amounts [32]. Similarly, *Gastridium geographus* injects three potent and paralytic ω-conotoxins, ω-GVIA, ω-GVIIA, and ω-GVIIB [28,33].

An unusual conotoxin, σ-GVIIIA, was identified in the defense venom of *Gastridium geographus* (Table 7) [28,33]. This toxin is a σS-conotoxin presenting a VIII cysteine framework and has been characterized as a blocker of the 5-HT_3_ serotonin receptor (“involved in the inhibition of neurotransmitter release at motor and sensory synapses“) [76]. Additional conopeptides, such as conophysin-G [28], were also identified in the defensive strategies. Interestingly, *Gastridium geographus* also employs two unclassified conotoxins G5.1 and a so-called “scratcher peptide”, which was named from the “scratching” symptoms observed in mice [77]. No data are available for the defensive strategies of other piscivorous clades, including *Chelyconus* and *Textilia*.

## 3. Molluscivorous Cone Snails

### 3.1. Predatory Venom

A little over 60 cone snails have been identified as mollusc-hunters, which represents a smaller portion over the fish- and worm-hunting species [31]. It is important to note that some cone snails do not exclusively prey on a single type of mollusc prey, but have adapted to more than one [31]. Although a complete genome assembly from mollusc-hunters is still lacking, it is fairly possible that mollusc-hunting cone snails have originated from a single root in opposition to piscivorous cone snails, which have been shown to be polytopic [31]. From Puillandre et al.’s phylogenetic classification, these mollusc-eating cone snails have been classified into six different clades: *Calibanus*, *Conus*, *Cylinder*, *Darioconus*, *Eugeniconus*, and *Leptoconus* [2].

The *Cylinder* and *Conus* clades represent the most studied in general, as they regroup some of the larger cone snails, such as the common *Cylinder textile* and *Conus marmoreus*. Fewer data are found in comparison to the extensively studied piscivorous cone snails, and even less on the predatory and defensive differentiation of venom for mollusc-hunters, limited so far to three species from the *Cylinder* clade (*ammiralis*, *textile*, and *victoriae)*, as well as one from the *Conus* clades (*marmoreus*) (Figure 10).

Compared to piscivorous species, the predatory-evoked venom of mollusc-hunters shows high complexity. Analyses of *Conus* and *Cylinder* species have permitted the identification of 17 gene superfamilies, 7 conopeptide classes, and a few unclassified toxins (Figure 11). Within this high diversity of conotoxins injected, M-, O1, T, and O2 superfamilies dominate the predatory venoms in general, followed by I1, A, I2, and H superfamilies. Additionally, a few conopeptide classes, such as insulins, contryphans, conophysins/conopressins, and conorfamides, were also identified, as well as less common classes, such as conomarphins, elevin, and even prohormones (Figure 11).

Although the venom of a single cone snail from the *Conus* clade was characterized (*Conus marmoreus*), it provided almost half of the knowledge gathered on the predatory strategy of mollusc-hunters. Indeed, the *Conus* clade predatory venom comprises 14 out of a total of 24 gene superfamilies. Meanwhile, the 3 *Cylinder* cone snails together yielded only slightly more diversity in their predatory venoms, with 20 gene superfamilies. From these, two gene superfamilies and types of conopeptides are found exclusively in the predatory venom of *Conus marmoreus* (respectively, B2 and I2, and conomarphins and contryphans), while *Cylinder* cone snails present exclusively five gene superfamilies and five types of conopeptide (F, P, R, elevin, and prohormone in *Cylinder victoriae* [28,78,79], and I4, J, conophysin/conopressin, conorfamide, and insulin in *Cylinder ammiralis* [80]) (Figure 11).

Superfamily A conotoxins are found in both molluscivorous clades investigated but they appear not as prevalent as in the piscivorous cone snails (Table 8). Some of these conotoxins have a similar structure and function as the α-conotoxins from fish-hunters, such as α-Mr1.1 (*Conus marmoreus*) [81] and α-VcIA (*Cylinder victoriae*) [28]. A very recent proteomic study has been conducted to uncover the predatory and defensive venoms of *Cylinder ammiralis*, which allowed the characterization of new sets of conotoxins and conopeptides, with some still unknown [80]. The predatory venom of *Cylinder ammiralis* shows α-like conotoxins, Ai1.2 and Ai1.2, but they have not been characterized yet [80].

As mentioned, M-conotoxins are of the most abundant in predatory venoms as well as O1- and T-conotoxins (Figure 11). M-conotoxins (Table 9), such as MrIIIB (*Conus marmoreus*) and TxIIIC (*Cylinder textile*), seemed to induce several symptoms in mice upon intracranial injection, such as scratching, hyperactivity, and circular motion [82], while Mr1e induced excitatory effects [83]. However, it is unknown if these effects reported in mammalian animal models can be translated to mollusc prey physiology.

O1-Conotoxins are also highly produced in the predatory venom of both clades. Most of them uphold the classical VI/VII framework (Table 10). Among them, we find μ-conotoxins such as μ-MrVIA and μ-MrVIB [81] from *Conus marmoreus*, which block Na^+^ currents of VGSCs [84]. The sequence homology can be clearly seen between M-conotoxins produced by the different species of mollusk-hunters, for example between TxO1 (*Cylinder textile*) [16] and P_019 (*Cylinder ammiralis*), with this conservation indicating a likely important ecological role [80].

A few O2-conotoxins were identified from the predatory venom of both *Cylinder ammiralis* and *Conus marmoreus*. These include a majority of almost 30 amino-acid long conotoxins, upholding a VI/VII cysteine framework. In addition, a contryphan-like conopeptide (P_163) was also identified in the venom of *Cylinder ammiralis* (Table 11) [80].

One other major family identified in the predation of molluscivorous cone snails is the T superfamily, which is absent from piscivorous species (Table 12). The T-conotoxins are smaller peptides that uphold a V or X framework made of four cysteine residues. Only a few χ-conotoxins have been identified, some of which inhibit the antidepressant binding site of the NE transporter (dependent on Na^+^), such as χ-MrIA [81], in *Conus marmoreus*. Interestingly, χ-MrIA was identified as an analgesic, efficacious in managing neuropathic pain in mice experiments [81].

Finally, in addition to the conotoxins, a few conopeptides were also identified, which include contryphans, conomarphins, a conoporin, a conorfamide, conopressin/conophysin, elevin, prohormone, insulin, and other unidentified toxins (Table 13). The contryphan-M was identified as having similar activity to ω-conotoxins [54,85].

Pharmacological characterization of conotoxins from mollusc-hunting cone snails remains far behind compared to piscivorous species, rendering the interpretation of their predatory strategies more difficult. Nonetheless, a few have been characterized, such as α-conotoxins α-Mr1.1 [80] and α-VcIA [28], μO-conotoxins such as μ-MrVIA and μ-MrVIB, and the δ-conotoxin, δ-TxVIA (Table 10). Unusual pharmacological families are also characterized in some predatory venoms, such as χ- and ε-conotoxins, including χ-MrIA [80], χ-MrIA [81], χ/λ-CMrX [82], and ε-TxVA [83] from the T-superfamily (Table 12), as well as the χ/λ-CMrVIA [82] and the ω-contryphan M [80] (Table 13).

### 3.2. Defense Venom

The defensive venom of molluscivorous cone snails has only been described for three species: *Conus marmoreus*, *Cylinder ammiralis*, and *Cylinder victoriae*. Similar to the predatory venoms, defensive venoms are complex, and a high diversity of conotoxin gene superfamilies are detected. A total of 21 gene superfamilies were identified: A, H, F, I1, I2, J, M, N, O1, O2, P, R, S, T, U, contryphan, conoporin, conodipine, conopressin/conophysin, conorfamide, and prohormones (Figure 12). The majority of the conotoxins recovered belong to M, O1, O2, and T superfamilies, which is reminiscent of the predatory venoms, although the nature of the individual conotoxin sequences is often different (Figure 11).

M-Conotoxins included a few toxins that were used in both predatory and defense venoms (Table 9), such as Mr1e, Mr3.8, MrIIIB, MrIIID, MrIIIE, MrIIIF, and MrIIIG in *Conus marmoreus*, as well as P_147 and P_063 in *Cylinder ammiralis* (Table 9). In the same way, O1- and T-conotoxins included toxins that are shared in both strategies, such as the μO-conotoxins, μ-MrVIA and μ-MrVIB, and VcVIB (Table 10), and the T-conotoxins, χ-MrIA and VcVA (Table 12). Some O2-conotoxins, only identified from the predatory venom of *Cylinder ammiralis*, included contryphan-like toxins, such as D_054 and P_163, which show similarities with contryphan-M (Table 11) [80]. Conopeptides are also found in the defensive venom, which include contryphan-M, and conoporin, conorfamide, conodipine, conopressin/conophysin, and prohormones (Table 13).

## 4. Vermivorous Cone Snails

### 4.1. Predatory Venom

While they present the highest number of species within the Conidae, vermivorous cone snails have the least amount of published data on their predatory and defensive venoms. Most likely because of the difficulty in collecting the injected venom (small radula tooth, specific type of prey worms, etc.), the investigation of vermivorous cone snails is lagging behind [31]. Consequently, this lack of investigation impacted the study of their predatory and defensive venoms. A few cone snails have been studied with that intent, such as the *Stephanoconus/Rhombiconus* cone snail, *Stephanoconus imperialis* [9], as well as two *Rhizoconus* cone snails, *Rhizoconus vexillum* and *Rhizoconus capitaneus* [86], and a single *Vituliconus* cone, *Vituliconus planorbis* [87] (Figure 13).

So far, from the worm-hunters, only the predatory venom of *Stephanoconus imperialis* could be collected. Its analysis has revealed a moderately complex venom, which presents 12 gene superfamilies (Figure 14) [9]. Interestingly, we observe conotoxins with different types of cysteine frameworks, which differ from the other types of cone snails (Table 14). Overall, the K-conotoxins were the most abundant in the venom of *Stephanoconus imperialis*. Indeed, three K-conotoxins, Im23a, Im23b, and Im23.4, were identified with a XXIII framework and a structure containing two helices [9]. Their biological activity is unknown. An α-conotoxin, Im1.1, was also identified.

### 4.2. Defense Venom

Defense venoms have been recorded from vermivorous *Rhizoconus* and *Vituliconus* cone snails (Figure 15). First, defense venom of the *Vituliconus* cone snail, *Vituliconus planorbis*, presented a set of A-, J-, M-, O2-, T-, Y-, and a new superfamily-1 (NSf-1). The most represented were NSf-1, A, and M gene superfamilies (Table 14) [87]. *Rhizoconus* worm-hunting cone snails appear to use massively and almost exclusively homodimeric αD-conotoxins for their defense strategies (Table 15). Indeed, αD-conotoxins, presenting 10 cysteines arranged in a unique XX framework, were discovered in the defensive-evoked venom of both *Rhizoconus vexillum* and *Rhizoconus capitaneus* [86]. They can be detected in their dimeric form between 10 and 12 kDa. Similar to some defense-related αA-conotoxins, they potently inhibit the α7 nAChR subtype in mammalian assays [86], but more information is needed to understand their defensive role [86].

## 5. Discussion

When first discovered, the remarkable ability of cone snails to produce different venoms for specific purposes (predation or defense) was totally unexpected. For the first time, within the same animal, was shown that a defensive injection was not simply equivalent to a predatory sting, and vice versa. While this novel paradigm would have immediate implications for the management of envenomation victims for instance, or for our understanding of cone snail biology, it also argues for the future need to characterize individual conotoxins on the correct animal model to avoid erroneous conclusions on their true ecological role. In the case of vertebrate venomous animals, such as snakes, both preys (often small vertebrate mammals, but also reptiles and birds) and predators (mostly higher vertebrates) share a very conserved physiology, explaining how by a fortunate coincidence, the same toxins would be effective both in capturing prey and in defending against a predator [88]. To the contrary, cone snails had to deal with more complicated venom uses (very diverse vertebrate/invertebrate preys and predators), stimulating the evolution of distinct strategies [89]. We, therefore, encourage future studies to investigate predatory venoms on laboratory animals more closely related to prey types, such as zebrafish (piscivorous), *Lymnaea* or *Aplysia* snails (molluscivorous), and any worms (vermivorous), including *Caenorhabditis elegans*, although annelids would be preferable over nematodes.

The controlled and selective injection of different conotoxin cocktails suggest that the evolution of cone snail venom is not only under the direct influence of one but at least two, potentially equally important, driving forces: predation and defense. Therefore, it provides a unique opportunity to study venom–ecology relationships in unprecedented details, on the condition that both predatory and defensive venoms can be collected and analyzed separately. From the current literature, it appears that injected venom can theoretically be collected from all cone snail diet groups. However, this review exposes a clear bias toward fish-hunting species in comparison to the other feeding groups. This bias likely has to do with the ease of “milking”, as piscivorous species tend to use the large and strong radula tooth, which facilitates the collection procedure. In addition, a fish prey can be conveniently obtained in most laboratories, as piscivorous cone snails will happily accept freshwater fish (goldfish) or zebrafish (with the caveat of ethical considerations around using vertebrate animals). Molluscivorous species also use a large radula tooth, which are thinner and more flexible, rendering the milking more technical but still relatively simple. The real challenge comes with the tiny radula tooth of most worm-hunting species (often <1 mm), which rarely pierces through both the prey tissue and the parafilm for successful milking (personal experiences).

An important note relates to the description of injected venoms prior to 2014 and the knowledge about distinct predatory and defensive venoms in cone snails. Indeed, some of the variations observed in venom composition in “milked venoms” could be *a posteriori* attributed to the non-discrimination between predatory and defensive behaviors. Another area of discrepancy concerns the study of individual vs. pooled injected venoms. From personal experience, pooling large batches of injected venom could also “falsely” increase venom complexity (accumulating all subtle individual variations in one complex lot) and, more importantly, could also lead to inadvertent mixing of defensive and predatory venoms. Therefore, for pharmacological characterization of conotoxins, larger amounts of venom are required, and pooling can be acceptable in this case, but investigation of precise envenomation strategies should be limited to individual milking. Lastly, the use of proteomic softwares that allow the automated interpretation of MS/MS spectra and identification of peptides and proteins from sequence databases may generate substantial amounts of false positives if not carefully curated [90]. Therefore, these could also artificially inflate the number of conotoxin sequences identified in injected venoms.

With these considerations and limitations in mind, a high diversity of conotoxin gene superfamilies was identified in the composition of predatory and defensive venoms to date (summarized in Figure 16). However, it is puzzling that particular species can inject a very restricted set of conotoxins selected from the complex repertoire present in their venom duct. For instance, the piscivorous species of the *Pionoconus* clade often rely on a very simple predatory venom composition to subdue fish prey, with sometimes only one class of conotoxin injected (κA-conotoxins). Similarly, some vermivorous species of the *Rhizoconus* clade have evolved a defensive strategy almost exclusively centered around the injection of αD-conotoxins. The conotoxin selection mechanisms are not understood, but likely under the control of the nervous and/or hormonal systems. Importantly, some species from more basal clades show a non-differentiated venom duct, suggesting that they may not have the ability to produce two types of venom, although this remains to be experimentally demonstrated [91].

Ideally, further works on cone snails should systematically include the characterization of predatory and defense venoms when possible. The more data collected on different clades and diet types, the closer we will come to truly understand the intended evolved venom use and the ecological role of each conotoxin. For instance, no data are available for the predatory venom of any *Textilia* species, which may seem surprising, considering the piscivorous diet, large radula tooth, and relatively wide geographical distribution of some species in this clade, such as *Textilia bullatus*. Moreover, the characterization of the defense venoms is still lacking for many cone snail clades, especially for common and large molluscivorous, such as *Cylinder textile*, *episcopatus*, *aulicus*, or *canonicus*, and other members of the molluscivorous *Conus* clade (e.g., *Conus bandanus* or *Conus araneosus*). Another future line of research should investigate the possible use of small molecules in either predatory or defensive (or both) venoms [92]. Indeed, recent publications have focused on the characterization of non-peptidic components in the venom of cone snails [93]. Some of these small molecules were even ascertained to have a direct role in the prey-capture strategy of a vermivorous species on the basis of their structural and functional resemblance to polychaete mating pheromones [94]. However, critically, the role of small molecules in the ecology of cone snails can only be considered relative to their presence (or absence) in the injected venoms.

With the advent of more and more sophisticated tools to investigate venom, including those in the field of mass spectrometry, peptide synthesis, and high-throughput sequencing and screenings, novel details about the predatory and defensive strategies in cone snails are likely to be revealed in the near future [95]. This has also allowed the uprising of multi-omics strategies known as “venomics” by integrating genomics, proteomics, and transcriptomics to accelerate the characterization of complex venoms [56]. For instance, it might be possible to test the hypothesis of the release of venom components in the water in the case of piscivorous net-hunting species. Furthermore, new AI (artificial intelligence)-based bioinformatic tools may also provide assistance in the interpretation of “big data”, in visualizing conotoxin–receptor interactions at atomic levels, or in developing novel hypotheses. However, the main challenge will remain uncovering the pharmacology of conotoxins through prey- and predator-relevant bioassays.

## Figures and Tables

**Figure 1 toxins-16-00094-f001:**
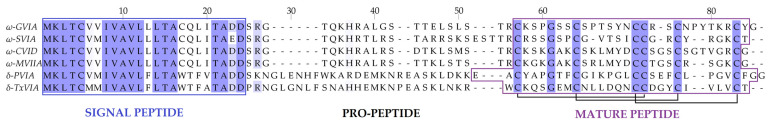
Conotoxin precursors. An alignment of six conotoxins belonging to the same gene superfamily (O1). The signal region (framed in blue) presents a sequence of highly conserved residues, mainly hydrophobic, while the mature region (framed in purple) presents more diversity of sequence and a greater number of cysteine residues. Conotoxin precursors: ω-GVIA (*Gastridium geographus*), ω-SVIA (*Pionoconus striatus*), ω-CVID (*Pionoconus catus*), ω-MVIIA (*Pionoconus magus*), δ-PVIA (*Chelyconus purpurascens*), and δ-TxVIA (*Cylinder textile*). The conotoxin precursors were aligned, amino acid residues were highlighted (in purple) according to the conservation, and disulfide bonds are represented with black lines.

**Figure 2 toxins-16-00094-f002:**
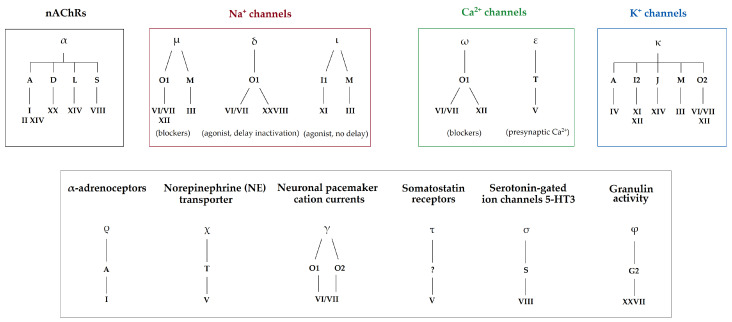
Pharmacological classification of conotoxins according to their gene superfamilies and cysteine framework. Pharmacological families are defined by Greek letters (α, μ, δ, ι, ω, ε, κ, ρ, χ, γ, τ, σ, and φ), gene superfamilies by Arabic capital letters (A, B, C, D, E, F, G, H, I, J, etc.), and cysteine frameworks by roman numbers (I, II, III, IV, V, VI, etc.). Identified biological targets may be linked to one or several pharmacological families (i.e., voltage-gated Na^+^ channels are targeted by µ-, δ-, and ι-conotoxins) [3,5,10].

**Figure 3 toxins-16-00094-f003:**
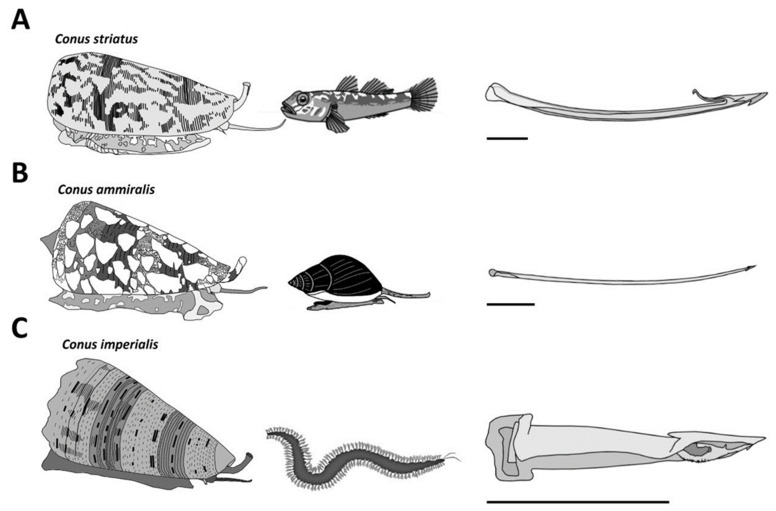
Major diet types observed in cone snails. (**A**) The piscivorous diet is represented here with a *Pionoconus striatus* specimen, which uses a “taser-and-tether” strategy to subdue its fish prey. The radula tooth is modified into a mini-harpoon. (**B**) *Cylinder ammiralis* is a molluscivorous species that injects thick venom multiple times through fine and long arrow-like radula teeth to incapacitate its gastropod prey. (**C**) *Stephanoconus imperialis*, which preys exclusively on amphinomid worms, uses a short and stout radula tooth to forcefully inject its greenish venom in large quantities. Horizontal bars indicate 1 mm.

**Figure 4 toxins-16-00094-f004:**
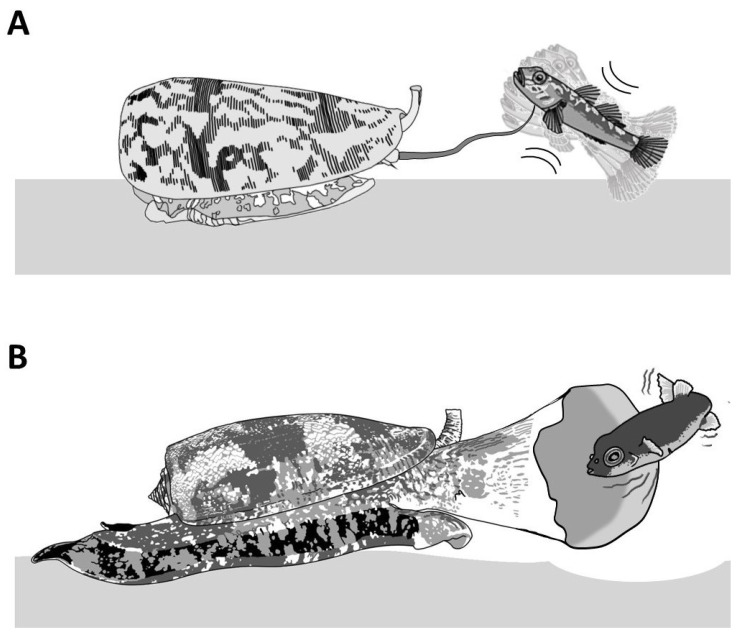
Piscivorous “taser-and-tether” and “net-hunting” strategies. (**A**) *Pionoconus striatus* is the prototypical species that uses a “taser-and-tether” strategy. The extended proboscis is reminiscent of a fish line and the radula tooth modified into a mini-harpoon to tether a prey. (**B**) The net-hunting strategy of a *Gastridium geographus* implies the extension of its rostrum in order to engulf a school of fish, which are already dazed by the hypothetical release of sedative compounds in the water.

**Figure 5 toxins-16-00094-f005:**
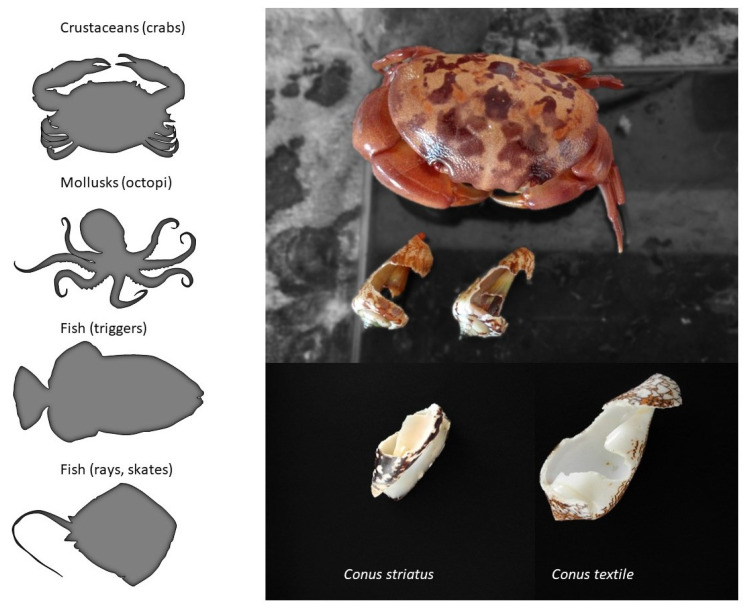
Natural predators of cone snails. The left panel shows the known predators of cone snails, whereas on the right is an example of the damages caused by a crab that was held in captivity together with various mollusks, including cone snails.

**Figure 6 toxins-16-00094-f006:**
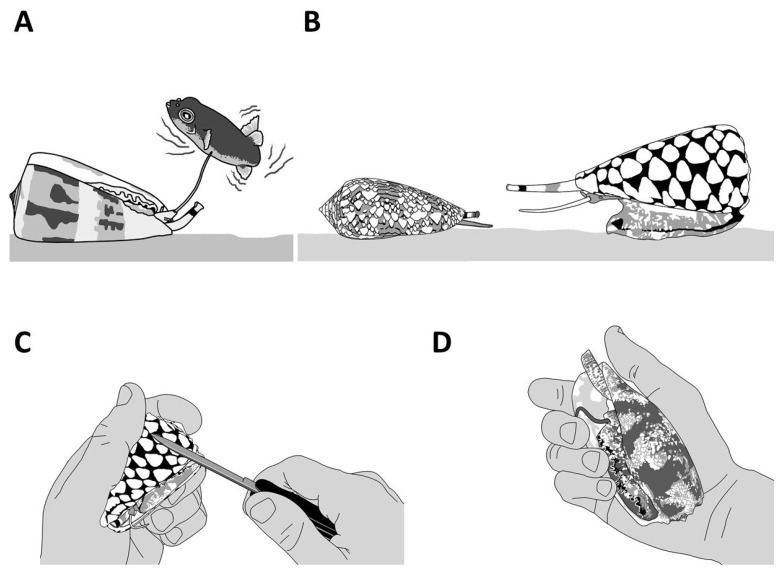
The defensive behaviors of cone snails. A defensive reaction can be triggered by different means, including using a natural predator (**A**,**B**) or aggravating the animal by directly interacting with it (**C**) or applying pressure to the shell (**D**). Live cone snails should not be handled.

**Figure 7 toxins-16-00094-f007:**
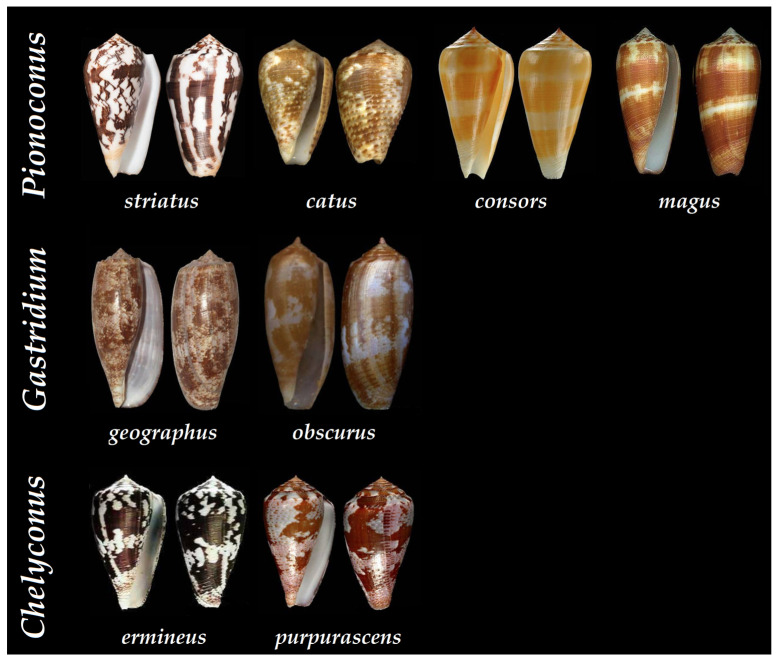
Shells of some of the piscivorous cone snails that have been characterized at the peptide level according to their clade. *Pionoconus striatus*, *Pionoconus catus*, *Pionoconus consors*, *Pionoconus magus*, *Gastridium geographus*, *Gastridium obscurus*, *Chelyconus purpurascens*, and *Chelyconus ermineus* [34].

**Figure 8 toxins-16-00094-f008:**
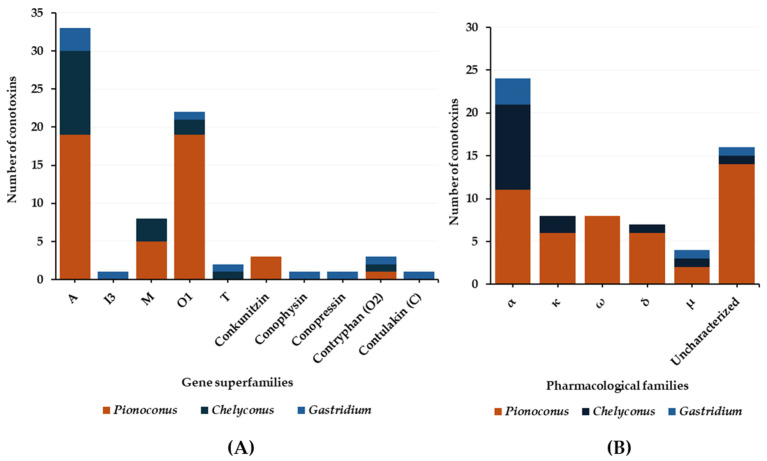
Gene superfamilies (**A**) and pharmacological families (**B**) identified within the predatory-evoked venoms of piscivorous cone snails.

**Figure 9 toxins-16-00094-f009:**
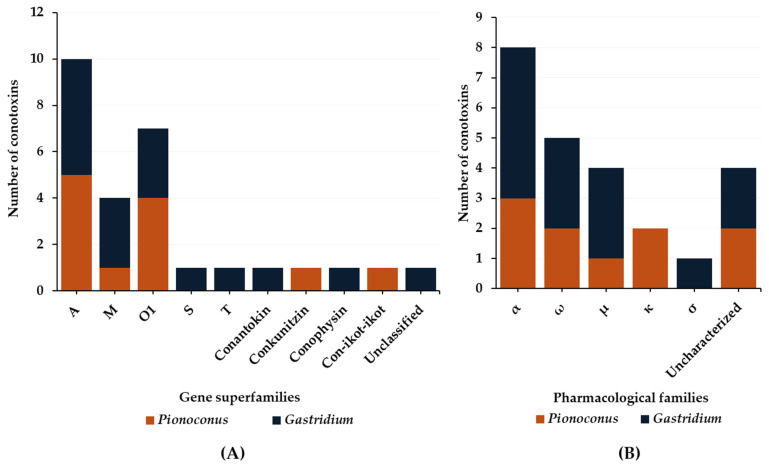
Gene superfamilies (**A**) and pharmacological families (**B**) identified within defense-evoked venoms of piscivorous cone snails. Cone snails include *Pionoconus striatus*, *Gastridium geographus*, and *Gastridium obscurus*.

**Figure 10 toxins-16-00094-f010:**
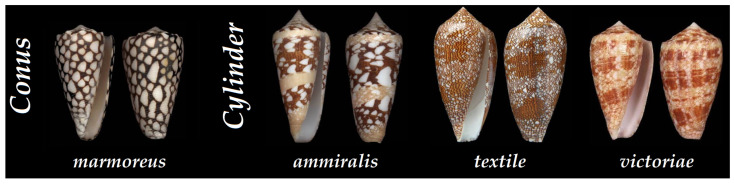
Shells of some of the molluscivorous cone snails that have been characterized at the peptide level according to their clade. *Conus marmoreus*, *Cylinder ammiralis*, *Cylinder textile*, and *Cylinder victoriae* [34].

**Figure 11 toxins-16-00094-f011:**
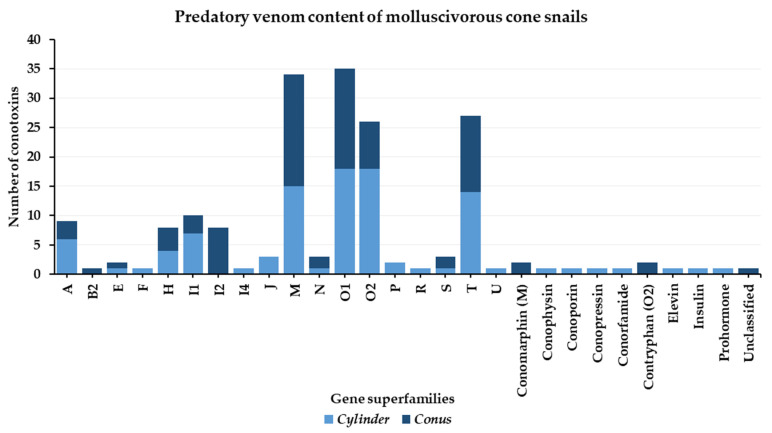
Gene superfamilies identified within predatory-evoked venoms of molluscivorous cone snails (*Conus marmoreus*, *Cylinder ammiralis*, *Cylinder textile*, and *Cylinder victoriae*).

**Figure 12 toxins-16-00094-f012:**
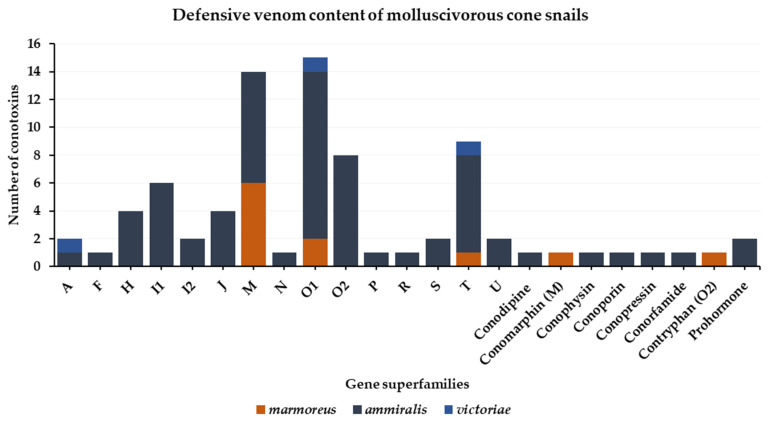
Gene superfamilies identified within defense-evoked venoms of molluscivorous cone snails (*Conus marmoreus*, *Cylinder ammiralis*, and *Cylinder victoriae*). No defense venoms were described from *Cylinder textile*.

**Figure 13 toxins-16-00094-f013:**
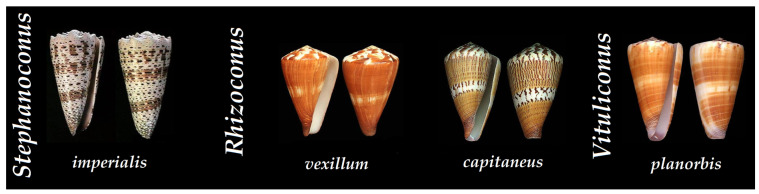
Shells of vermivorous cone snails that have been characterized at the peptide level according to their clade. *Stephanoconus imperialis*, *Rhizoconus vexillum*, *Rhizoconus capitaneus*, and *Vituliconus planorbis* [34].

**Figure 14 toxins-16-00094-f014:**
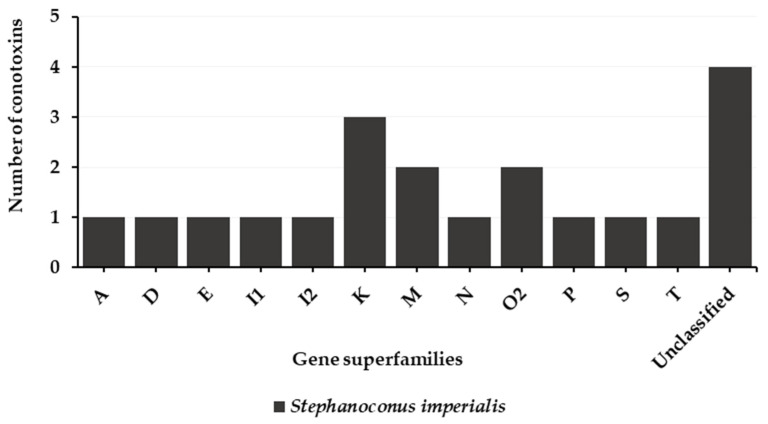
Gene superfamilies identified within predatory-evoked venoms of a vermivorous cone snail (*Conus imperialis* (*Stephanoconus*)) [9]. So far, this is the only predatory venom isolated from a worm-hunting cone snail.

**Figure 15 toxins-16-00094-f015:**
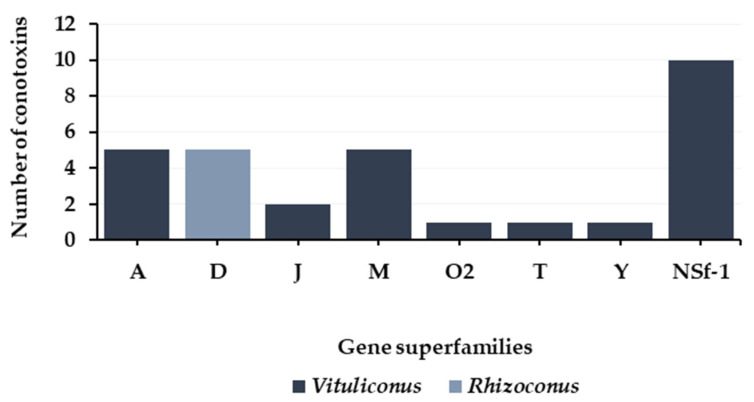
Gene superfamilies identified within defense-evoked venoms of vermivorous cone snails. *Conus vexillum*, *Conus capitaneus* (*Rhizoconus*) [86], and *Conus planorbis* (*Vituliconus*) [87]. NSf-1: new superfamily-1, a new gene superfamily that was identified in the venom of *Vituliconus planorbis* [87].

**Figure 16 toxins-16-00094-f016:**
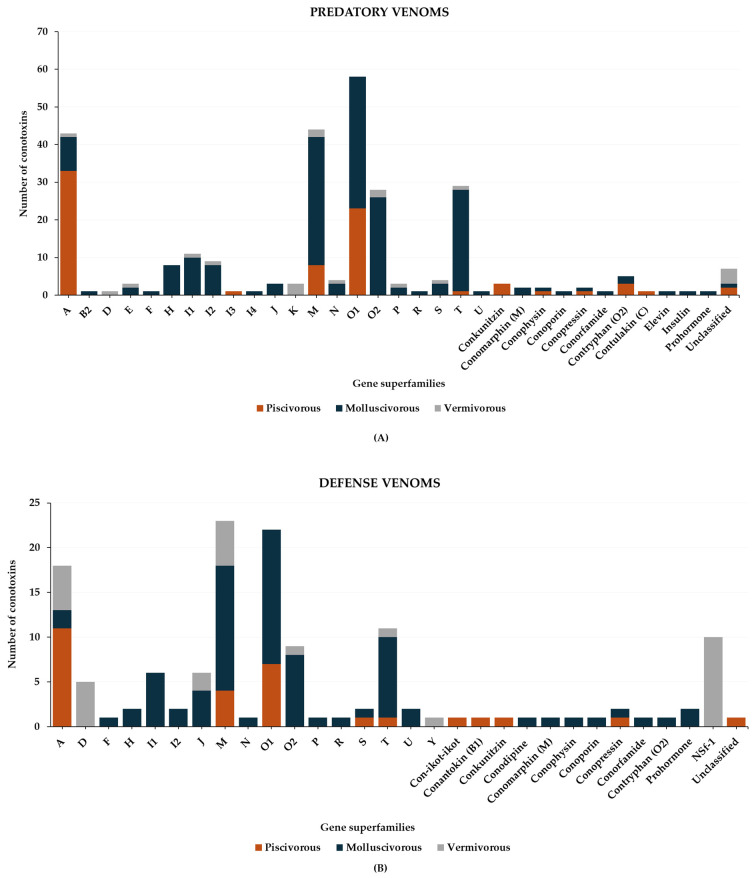
Summary of the gene superfamilies identified in the predatory (**A**) and defense venoms (**B**) of piscivorous, molluscivorous, and vermivorous cone snails belonging to different clades: *Pionoconus*, *Chelyconus*, *Gastridium*, *Cylinder*, *Conus*, *Stephanoconus*, *Rhizoconus*, and *Vituliconus*.

**Table 1 toxins-16-00094-t001:** κ-Conotoxins identified from predatory and defense venoms of fish-hunting cone snails. Presented here are conotoxins found exclusively in the predation-evoked or in both venoms. Each conotoxin is characterized by its *Conus* clade, the *Conus* species in which it was detected, the given name, its sequence, its classification within the gene superfamilies, and the cysteine framework. Cysteine residues are highlighted in **red**.

Clades	*Conus* Species	Conotoxins	Mature Sequence	Gene Superfamily	Cysteine Framework	References
*Pionoconus*	*striatus*	κA-SIVA	ZKSLVP(gSr)VITTCCGYDOGTMCOOCRCTNSCX	A	IV	[32]
		κA-SIVB	ZKELVP(gSr)VITTCCGYDOGTMCOOCRCTNSCOTKOKKOX	A	IV	[32]
		κA-SIVC	AOAL(I)VVTATTNCCGYTGOACHOCL(I)CTQTC		IV	[36]
	*catus*	C4.41	QKELVPSTITTCCGHEPGTMCPKCMCDNTCPPQKEEKTRPQ	A	IV	[25]
		C1.5	QKELVPSTITTCCGNGTGDNVDPKCMCDNTSSPKKKKRP	A	I	[25]
	*consors*	κA-CcTx	AOWLVP(gSr)QITTCCGYNOGTMCOSCMCTNTC	A	IV	[24,37]
*Chelyconus*	*purpurascens*	κA-PIVE	DCCGVKLEMCHPCLCDNSCKNYGKX	A	IV	[17,38]
		κA-PIVF	DCCGVKLEMCHPCLCDNSCKKSGKX	A	IV	[17,38]

**Table 2 toxins-16-00094-t002:** α-Conotoxins identified from predatory and defense venoms of fish-hunting cone snails. Presented here are conotoxins found exclusively in the predation-evoked, the defense-evoked, or in both venoms. Each conotoxin is characterized by its *Conus* clade, the *Conus* species in which it was detected, the given name, its sequence, its classification within the gene superfamilies, and the cysteine framework. Cysteine residues are highlighted in **red**.

Clades	*Conus* Species	Conotoxins	Mature Sequence	Gene Superfamily	Cysteine Framework	References
*Pionoconus*	*striatus*	α-SI	ICCNPACGPKYSCX	A	I	[23,32,39]
		α-SIA	YCCHPACGKNFDCX	A	I	[23,32]
		α-SII	GCCCNPACGPNYGCGTSCS	A	II	[23,32,39]
	*consors*	α-CnIB	CCHPACGKYYSCX	A	I	[24,37]
		α-CnIA	GRCCHPACGKYYSCX	A	I	[24,37]
		CnIG	CCHPACGKYFKCX		I	[37]
		CnIJ	GRCCHPACGGKYFKCX	A	I	[37]
		CnIH	NGRCCHPACGKHFSCX	A	I	[37]
		CnIK	NGRCCHPACGKYYSCX	A	I	[37]
		CnIL	DGRCCHPACGKYYSCX	A	I	[37]
	*catus*	α-C4.3	NGRCCHPACGKHFSC	A	I	[25]
		α-CIB	GCCSNPVCHLEHPNACX	A	I	[25]
		α-C1.3	GCCSNPVCHLEHSNLCX	A	I	[25]
	*magus*	α-MI	GRCCHPACGKNYSCX	A	I	[40]
		α-MII	GCCSNPVCHLEHSNLCX	A	I	[40]
		α-MIC	CCHPACGKNYSCX	A	I	[40]
*Gastridium*	*geographus*	α-GIC	GCCSHPACAGNNQHICX	A	I	[28,33]
		α-GIA	ECCHPACGRHYSCGK	A	I	[28,33]
		α-GII	ECCHPACGKHFSCX	A	I	[28,33]
		α-GID	IRD(Gla)CCSNPACRVNNPHVC	A	I	[28,33]
	*obscurus*	α-OIVA	CCGVONAACHOCVCKNTCX	A	IV	[28,41]
		α-OIVB	CCGVONAACPOCVCNKTCGX	A	IV	[28,42]
*Chelyconus*	*purpurascens*	α-PIB	ZSOGCCWNPACVKNRCX	A	I	[17,43]
		α-PIC	SGCCKHOACGKNRC	A	I	[17,44]
		α-PIVA	GCCGSYONAACHOCSCKDROSYCGQX	A	IV	[17]
		α-PIIIE	HOOCCLYGKCRRYPGCSSASCCQRX	M	III	[17]
		α-PIIIF	GOOCCLYGSCROFOGCYNALCCRKX	M	III	[17,45]
	*ermineus*	α-EIVA	GCCGPYONAACHOCGCKVGROOYCDROSGGX	A	IV	[46,47]
		α-EIVB	GCCGKYONAACHOCGCTVGROOYCDROSGGX	A	IV	[46,47]
		α-EIIA	ZTOGCCWNPACVKNRCX	A	I	[47,48]
		α-EIIB	ZTOGCCWHPACGKNRCX	A	I	[47,48]
		α-EI	RDOCCYHPTCNMSNPQICX	A	I	[47,49]

**Table 3 toxins-16-00094-t003:** δ-Conotoxins identified from predatory and defense venoms of fish-hunting cone snails. Presented here are conotoxins found exclusively in the predation-evoked venoms. Each conotoxin is characterized by its *Conus* clade, the *Conus* species in which it was detected, the given name, its sequence, its classification within the gene superfamilies, and the cysteine framework. Cysteine residues are highlighted in **red**.

Clades	*Conus* Species	Conotoxins	Mature Sequence	Gene Superfamily	Cysteine Framework	References
*Pionoconus*	*consors*	δ-CnVIA	YECYSTGTFCGINGGLCCSNLCLFFVCLTFS	O1	VI/VII	[37]
		δ-CnVIB	DECFSOGTFCGTKOGLCCSARCFSFFCISLEFX	O1	VI/VII	[37]
		δ-CnVIC	DECFSOGTFCGIKOGLCCSARCFSFFCISLEFX	O1	VI/VII	[37]
	*striatus*	δ-SVIE	DGCSSGGTFCGIHOGLCCSEFCFLWCITFID	O1	VI/VII	[32]
	*catus*	δ-CVIE-2	YGCSNAGAFCGIHOGLCCSELCLVWCT	O1	VI/VII	[25]
		δ-C6.2	DGCYNAGTFCGIROGLCCSEFCFLWCITFVDSX	O1	VI/VII	[25]
*Chelyconus*	*purpurascens*	δ-PVIA	EACYAPGTFCGIKPGLCCSEFCLPGVCFGX	O1	VI/VII	[55,56]

**Table 4 toxins-16-00094-t004:** μ-Conotoxins identified from predatory and defense venoms of fish-hunting cone snails. Presented here are conotoxins found exclusively in the predation-evoked or the defense-evoked venoms. Each conotoxin is characterized by its *Conus* clade, the *Conus* species in which it was detected, the given name, its sequence, its classification within the gene superfamilies, and the cysteine framework. Cysteine residues are highlighted in **red**.

Clades	*Conus* Species	Conotoxins	Mature Sequence	Gene Superfamily	Cysteine Framework	References
*Pionoconus*	*consors*	μ-CnIIIB	ZGCCGEPNLCFTRWCRNNARCCRQQ	M	III	[24,37]
		μ-CnIIIC	ZGCCNGPKGCSSKWCRDHARCCX	M	III	[24,37]
	*striatus*	μ-S3-G02	QKCCGEGSSCPKYFKNNFICGCC	M	III	[32]
*Chelyconus*	*purpurascens*	μ-PIIIA	ZRLCCGFOKSCRSRQCKOHRCC	M	III	[56]
*Gastridium*	*geographus*	μ-Conotoxin-GS	ACSGRGSRCOOQCCMGLRCGRGNPQKCIGAH(Gla)DV	O1	VI/VII	[28]
		μ-GIIIA	RDCCTOOKKCKDRQCKOQRCCAX	M	III	[28,33]
		μ-GIIIB	RDCCTOORKCKDRRCKOMKCCAX	M	III	[33]
		μ-GIIIC	RDCCTOOKKCKDRRCKOLKCCA	M	III	[28,33]

**Table 5 toxins-16-00094-t005:** ω-Conotoxins identified from predatory and defense venoms of fish-hunting cone snails. Presented here are conotoxins found exclusively in the predation-evoked, the defense-evoked, or in both venoms. Each conotoxin is characterized by its *Conus* clade, the *Conus* species in which it was detected, the given name, its sequence, its classification within the gene superfamilies, and the cysteine framework. Cysteine residues are highlighted in **red**.

Clades	*Conus* Species	Conotoxins	Mature Sequence	Gene Superfamily	Cysteine Framework	References
*Pionoconus*	*striatus*	ω-SVIA	CRSSGSOCGVTSICCGRCYRGKCTX	O1	VI/VII	[39,64]
		ω-SVIB	CKLKGQSCRKTSYDCCSGSCGRSGKCX	O1	VI/VII	[39,64]
		SO4	ATDCIEAGNYCGPTVMKICCGFCSPYSKICMNYPKN	O1	VI/VII	[23,32]
		SO5	STSCMEAGSYCGSTTRICCGYCAYFGKKCIDYPSN	O1	VI/VII	[23,32]
	*consors*	ω-CnVIIA	CKGKGAOCTRL(Mox)YDCCHGSCSSSKGRCX	O1	VI/VII	[24,37,51]
	*magus*	ω-MVIIA	CKGKGAKCSRLMYDCCTGSCRSGKCX	O1	VI/VII	[40]
		ω-MVIIB	CKGKGASCHRTSYDCCTGSCNRGKCX	O1	VI/VII	[40]
	*catus*	ω-Catus-C2	CQGRGASCRKTMYNCCSGSCNRGRC	O1	VI/VII	[25]
		ω-CVIA	CKSTGASCRRTSYDCCTGSCRSGRCX	O1	VI/VII	[25,61]
		ω-CVID	CKSKGAKCSKLMYDCCSGSCSGTVGRCX	O1	VI/VII	[25,61]
*Gastridium*	*geographus*	ω-GVIA	CKSOGSSCSOTSYNCCRSCNOYTKRCY	O1	VI/VII	[28,33]
		ω-GVIIA	CKSOGTOCSRGMRDCCTSCLLYSNKCRRY	O1	VI/VII	[28,33]
		ω-GVIIB	CKSOGTOCSRGMRDCCTSCLSYSNKCRRY	O1	VI/VII	[28,33]

**Table 6 toxins-16-00094-t006:** Conopeptides identified from predatory and defense venoms of fish-hunting cone snails. Presented here are conotoxins found exclusively in the predation-evoked, the defense-evoked, or in both venoms. Each conotoxin is characterized by its *Conus* clade, the *Conus* species in which it was detected, the given name, its sequence, its classification within the gene superfamilies, and the cysteine framework. Cysteine residues are highlighted in **red**.

Clades	*Conus* Species	Conotoxins	Mature Sequence	Gene Superfamily	Cysteine Framework	References
*Pionoconus*	*striatus*	Conkunitzin S1	KDRPSLCDLPADSGSGTKAEKRIYYNSARKQCLRFDYTGQGGNENNFRRTYDCQRTCLYT	Conkunitzin	XIV	[32]
		Str54	PSYCNLPADSGSGTKPEQRIYYNSAKKQCVTFTYNGKGGNGNNFSRTNDCRQTCQYPLYACISGCRCET	Conkunitzin		[32]
		Con-ikot-ikot S	SGPADCCRMKECCTDRVNECLQRYSGREDKFVSFCYQEATVTCGSFNEIVGCCYGYQMCMIRVVKPNSLSGAHEACKTVSCGNPCA	Con-ikot-ikot		[32]
		Conkunitzin S2	ARPKDRPSYCNLPADSGSGTKPEQRIYYNSAKKQCVTFTYNGKGGNGNNFSRTNDCRQTCQYPVG	Conkunitzin	XIV	[32]
*Chelyconus*	*purpurascens*	Contryphan-P	GCVLLPWC	O2		[35]
*Gastridium*	*geographus*	Contryphan-G	GCPWEPWC	O2		[32]
		Conopressin-G	CFIRNCPKGX	Conopressin		[28,33]
		Contulakin-G	QSEEGGSNATKKPYIL	C		[33]
		G5.1	QGWCCKENIACCV	T	V	[28,33]
		Scratcher peptide	KFLSGGFKIVCHRYCAKGIAKEFCNCPD		XIV	[28,33]
		Conantokin-G	GE(Gla)(Gla)LQ(Gla)NQ(Gla)LIR(Gla)KSN	B1		[28,33]
		Conophysin-G	THPCMSCSFGQCVGPQICCGLGGCEMGTAEANKCIEEDDDQTPCQVLGDHCDLNNLDIEGHCVADGICCVDDTCAIHSSC	Conophysin		[28]

**Table 7 toxins-16-00094-t007:** σ-Conotoxin identified from predatory venoms of a fish-hunting cone snail (*Gastridium geographus*). Presented here are conotoxins found exclusively in the defense-evoked venom. Each conotoxin is characterized by its *Conus* clade, the *Conus* species in which it was detected, the given name, its sequence, its classification within the gene superfamilies, and the cysteine framework. Cysteine residues are highlighted in **red**.

Clades	*Conus* Species	Conotoxins	Mature Sequence	Gene Superfamily	Cysteine Framework	References
*Gastridium*	*geographus*	σ-GVIIIA	GCTRTCGGOKCTGTCTCTNSSKCGCRYNVHPSG(Btr)GCGCACSX	S	VIII	[28,33]

**Table 8 toxins-16-00094-t008:** A-conotoxins identified from predatory and defense venoms of mollusc-hunting cone snails. Presented here are conotoxins found exclusively in the predation-evoked or in both venoms. Each conotoxin is characterized by its *Conus* clade, the *Conus* species in which it was detected, the given name, its sequence, and the cysteine framework. Cysteine residues are highlighted in **red**.

Clades	*Conus* Species	Conotoxins	Mature Sequence	Cysteine Framework	References
*Conus*	*marmoreus*	α-Mr1.1	GCCSHPACSVNNPDICX	I	[81]
		Mr1.8a	ECCTHPACHVSNPELCX	I	[81]
		Mr1.8	ROECCTHOACHVSNPELCS	I	[81]
*Cylinder*	*victoriae*	α-VcIA	GCCSDPRCNYDHPEICX	I	[28]
	*ammiralis*	Ai1.2	PECCSDPRCNSTHPELCG	I	[80]
		Ai1.1	QECCSYPACNLDHPELC	I	[80]
		P_113	AGINDVCKSWRDCPQGADCYVDVGLRCRWPSDHSCTANNQCSVDSCINGICKANIGGRCLSDRDCPKGATCKSQE	VIII	[80]
		P_165	DCPVTGGPNPYHHCMIACMADGTKEYCRCHYCKDCVDSNGDKPAC	XXII	[80]
		P_114	AGINDVCKSWRDCPQGADCYVDVGLRCRWPSDHSCTANNQCSVDSCINGICKANIGGRCLSNKDCPEGATCKSQGWLNFEKKCET	XXXIII	[80]

**Table 9 toxins-16-00094-t009:** M-conotoxins identified from predatory and defense venoms of mollusc-hunting cone snails. Presented here are conotoxins found exclusively in the predation-evoked or in both predatory and defense venoms. Cysteine residues are highlighted in **red**. The complete table can be found in the Appendix A.

Clades	*Conus* Species	Conotoxins	Mature Sequence	Cysteine Framework	References
*Conus*	*marmoreus*	Mr1.9	VCCPFGGCHELCTADDX	I	[81]
		Mr3.11	CCRIACNLKCNOCCX	III	[81]
		Mr3.15	VCCPHGGCHQICQCCGC	III	[81]
		Mr3.18	CCHRNWCDHLCSCCGS	III	[81]
		MrIIIF	VCCPFGGCHELCLCCDX	III	[81]
		Mr1e	CCHSSWCKHLC	I	[28,33,81]
		Mr3.8	CCHWNWCDHLCSCCGS	III	[81]
		MrIIIB	SKQCCHLAACRFGCTOCCW	III	[81]
		MrIIID	CCRLSCGLGCHOCCX	III	[81]
		MrIIIE	VCCPFGGCHELCYCCDX	III	[81]
		MrIIIG	DCCOLPACPFGCNOCCX	III	[81]
*Cylinder*	*textile*	TxIIIC	CCRTCFGCTOCCX	III	[16]
		Tx3f	RCCKFPCPDSCRYLCCX	III	[16]
		Tx3a	CCSWDVCDHPSCTCCGX	III	[16]
		Tx3h	KFCCDSNWCHISDCECCYX	III	[16]
	*ammiralis*	P_148	RCCRFPCPDTCRHLCC	III	[80]
		P_099	FCCDSDWCHLPECLCCN	III	[80]
		P_147	CCMTCFGCTPCC	III	[80]
		P_063	CCSWDVCDHPSCTCCS	III	[80]
		P_122	CCNDSECDYSCWPCCIFS	III	[80]
		P_143	CCSWDVCDHPSCTCC	III	[80]
		P_149	CCNAGFCRFGCTPCCWMTSFVIAASSSV	III	[80]
		P_151	VCCPFGGCHELCQCCE	III	[80]
		P_170	GILLPALRKFCCDSNWCHISDCECCY	III	[80]

**Table 10 toxins-16-00094-t010:** O1-conotoxins identified from predatory-evoked venoms of mollusc-hunting cone snails. Presented here are conotoxins found exclusively in the predation-evoked, the defense-evoked, or in both venoms. Each conotoxin is characterized by its *Conus* clade, the *Conus* species in which it was detected, the given name, its sequence, and the cysteine framework. Cysteine residues are highlighted in **red**. The complete table can be found in the Appendix A.

Clades	*Conus* Species	Conotoxins	Mature Sequence	Cysteine Framework	References
*Conus*	*marmoreus*	Mr6.22	CIDGGEMCDPFSSDCCSGWCIFFFCT	VI/VII	[81]
		Mr6.8	CIDGGEICDIFFPNCCSGWCIILVCA	VI/VII	[81]
		MaIr332	CLDGGEICGILFPSCCSGWCIVLVCA	VI/VII	[81]
		MaIr34	ECLEADYYCVLPFVGNGMCCSGICVFVCIAQKY	VI/VII	[81]
		MaIr137	DDECEPPGDFCGFFKIGPPCCSGWCFLWCA	VI/VII	[81]
		Mr6.17	ACRQKWEYCIVPILGFVYCCPGLICGPFVCV	VI/VII	[81]
		μ-MrVIA	ACRKKWEYCIVPIIGFIYCCPGLICGPFVCV	VI/VII	[81]
		μ-MrVIB	ACSKKWEYCIVPILGFVYCCPGLICGPFVCV	VI/VII	[81]
*Cylinder*	*textile*	TxO1	CLDAGEVCDIFFPTCCGYCILLFCA	VI/VII	[16]
		δ-TxVIA	WCKQSGEMCNLLDQNCCDGYCIVLVCT	VI/VII	[16]
		TxO4	YDCEPPGNFCGMIKIGPOCCSG(Btr)CFFACA	VI/VII	[16]
	*victoriae*	VcVIB	GKPCHEEGQLCDPFLQNCCLGWNCVFVCI	VI/VII	[28]
	*ammiralis*	P_020	CVDQFDPCDMIRHTCCVGVCFLMACI	VI/VII	[80]
		D_045	CKQADEPCSILSLDQCCSGVCFGICI	VI/VII	[80]
		D_050	ECQEKWDYCPIPFFGSRYCCYGLFCTLFFCA	VI/VII	[80]
		D_047	WCKQSGEMCNFTDQNCCDGYCILLFCT	VI/VII	[80]
		P_019	CTQSGELCDVIDPDCCNKFCIIFFCI	VI/VII	[80]
		P_162	CYDGGTSCNTGNQCCSGWCIFVCL	VI/VII	[80]
		P_076	VKPCRKEGQLCDPIFQNCCRGWNCVFVCI	VI/VII	[80]
		P_121	DDCEPPGNFCGMIKIGPPCCSGWCFFACA	VI/VII	[80]
		Ai6.1	WCKQSGEMCNLLDQNCCEGYCIVLVCT	VI/VII	[80]

**Table 11 toxins-16-00094-t011:** O2-conotoxins identified from predatory and defense venoms of mollusc-hunting cone snails. Presented here are conotoxins found exclusively in the predation-evoked, the defense-evoked, or in both venoms. Cysteine residues are highlighted in **red**. The complete table can be found in the Appendix A.

Clades	*Conus* Species	Conotoxins	Mature Sequence	Cysteine Framework	References
*Conus*	*marmoreus*	Mr6.13	DCLPIGSLCHSSEQCCSGWCSPKRVC	VI/VII	[28,81]
		Mr6.14	SCDQTGEPCVLNEQCCYGWCTNHGTCY	VI/VII	[28,81]
		Mr6.15	SCVPIGRPCASNEQCCTRWCTPRRIC	VI/VII	[81]
		Mr6.12	GCKATWMSCSSGWECCSMSCDMYCX	VI/VII	[81]
		MaI51	QCEDVWMPCTSNWECCSLDCEMYCTQIX	VI/VII	[81]
*Cylinder*	*ammiralis*	P_053	LCPDYTEPCSHAHECCSWNCHNGHCT	VI/VII	[80]
		P_167	NCSDDWQYCESPSDCCSWDCDVVCS	VI/VII	[80]
		D_054	CRMTPLC		[80]
		P_163	ECPWKPWC		[80]
		P_067	WWDGDCRTWNAPCNPGVECCFGRCSHRRCVFW	VI/VII	[80]

**Table 12 toxins-16-00094-t012:** T-conotoxins identified from predatory and defense venoms of mollusc-hunting cone snails. Presented here are conotoxins found exclusively in the predation-evoked or both predatory- and defense-evoked venoms. Cysteine residues are highlighted in **red**. The complete table can be found in the Appendix A.

Clades	*Conus* Species	Conotoxins	Mature Sequence	Cysteine Framework	References
*Conus*	*marmoreus*	MrVA	NACCIVRQCC	V	[81]
		Mr5.6	NGCCRAGDCCS	V	[81]
		χ/λ-CMrX	GICCGVSFCYPC		[81]
		Mr10.2	ACCVYKICYPC	X	[81]
		Gla-MrIII	FCCRTQ(Gla)VCC(Gla)AIKNX	V	[81]
		Mr5.1b	CCPGWELCC(Gla)WDDGW	V	[28,81]
		Mr5.4a	CCQVMPQCCEWN	V	[81]
		Mr5.4b	CCQIVPQCCEWN	V	[81]
		Mr5.8	CCQIVPQCCEWVSD	V	[81]
		χ-MrIA	NGVCCGYKLCHPC	X	[81]
*Cylinder*	*textile*	Tx-D0111	QCCWYFDISCCITV	V	[16]
		TxXIIIA	TSDCCFYHNCCC	V	[16]
	*victoriae*	VcVA	CCPGKPCCRIX	V	[28]
		Vc5.3	VNCCGIDESCCS	V	[28]
	*ammiralis*	P_175	RCCSIHDNSCCGL	V	[80]
		P_071	NMCCGFKPYCCNW	V	[80]
		P_038	HDMPLASFHGNAMRILQMLSNNRYCCIFDHSCCLWP	V	[80]
		P_159	SGCCVIDRNCC	V	[80]
		P_070	TSDCCFYHNCCC	V	[80]
		P_166	GCCSYFDVSCCLWP	V	[80]
		P_037	PCCSIHDSSCCGL	V	[80]
		P_039	FCCRPVTPCCA	V	[80]
		P_072	QACCGFKMCVPC	I	[80]
		P_097	NLQILCCKHTLSCCT	V	[80]

**Table 13 toxins-16-00094-t013:** Conopeptides identified from predatory and defense venoms of mollusc-hunting cone snails. Presented here are conotoxins found exclusively in the predation-evoked, the defense-evoked, or in both venoms. Cysteine residues are highlighted in **red**. The complete table can be found in the Appendix A.

Clades	*Conus* Species	Conotoxin/ Conopeptide	Mature Sequence	Conopeptide Class	Cysteine Framework	References
*Conus*	*marmoreus*	χ/λ-CMrVIA	VCCGYKLCHPC		X	[81]
		Contryphan-M2	ESECPWHPWCX	Contryphan		[81]
		ω-Contryphan-M	NESECPWHPWCX	Contryphan		[81]
		Conomarphin-Mr1	DWEYHAHPKPNSFWT	Conomarphin		[81]
		Conomarphin-Mr2	DWVNHAHPQPNSIWS	Conomarphin		[81]
*Cylinder*	*textile*	Textile Convulsant Peptide (TCP)	NCPYCVVYCCPPAYCEASGCRPPX		O1	[16]
	*victoriae*	Elevenin-Vc1	RRIDCKVFVFAPICRGVAA	Elevenin		[21]
		Prohormone-4-Vc1	IGFPGFSTPPR	Prohormone		[21]
		P_088	TTVEKNKPGVLDIPVKSNSDDDSIFRYGRRDMQSPLLSERLRF	Conorfamide		[80]
		P_087	SFGGEHVCWLGDPNHPQGVCGPQVANIVEIRCEEKEAEQGGANNARANTGRTSSLMKRRGFLSLLKKRGKRDEGSPLQRSGRGIVCECCKHHCTKEEFTEYCH	Insulin	XII	[80]
	*ammiralis*	D_086	HSGILLAWSGPRNRFVRFG	Conorfamide		[80]
		D_002	AQDYSTAELCRINSNDCSVPFSWIPCQQHFLAACDRHDTCYLCGAHFNFTQDDCDNAFLRDMTALCANGTDDEGFCLQ	Conodipine	XXXIII	[80]
		D_030	ASCPENSCDFASPFQCGEMQTCIRLFQVCDGLWHCENGFDEDLAVCAAVLRPLECAIWEFLEEQSDWILPELFNNADSDLVAPVLHGAYSMGDLQSILNLTAQNIENIRNSTRGAIEGDERPLLALGMPEGAWNDVRYLLEELYKLGLDVWTE	Prohormone-4	XXII	[80]
	*ammiralis*	P_016	INCKVFVYAPICRGVAA	Conoporin		[80]
		P_158	HPTKACMNCTFGQCVGPQVCCGAGGCEMGTAEANRCSEEDEDPIPCLVIGAHCSLNNPGNIHGNCVAHGICCVDDTCAIHFGCL	Conophysin		[80]
			CFIRNCPSGG	Conopressin		[80]

**Table 14 toxins-16-00094-t014:** Various conotoxins identified from predatory and defense venoms of worm-hunting cone snails. Presented here are conotoxins found exclusively in the predation-evoked or the defense-evoked venoms. Cysteine residues are highlighted in **red**. The complete table can be found in the Appendix A.

Clades	*Conus* Species	Conotoxins	Mature Sequence	Gene Superfamily	Cysteine Framework	References
*Stephanoconus*	*imperialis*	Im1.1	DYCCHRGPCMVWC	A	I	[9]
		Im22.1	NCKKNILRTYCSNKICGEATKNTNGELQCTMYCRCANGCFRGQYIDWPNQQTNLLFC	E	XXIII	[9]
		Im11.8	CSDNIGATCSDRFDCCGSMCCIGGQCVVTFAECS	I1	XI	[9]
		Im11.9	CHMDCSKMTCCSGICCFYCGRPMCPGT	I2	XI	[9]
		Im23b	IPYCGQTGAECYSWCIKQDLSKDWCCDFVKTIARLPPAHICSQ	K	XXIII	[9]
		Im23.4	VPYCGQTGAECYSWCKEQHLIRCCDFVKYVGMNPPADKC	K	XXIII	[9]
		Im23a	IPYCGQTGAECYSWCIKQDLSKDWCCDFVKDIRMNPPADKCP	K	XXIII	[9]
*Vituliconus*	*planorbis*	Pl1.1	GIRGNCCMFHTCPIDYSRFYCP	A	I	[87]
		Pl169	TVIMHNCCTRSFCKRIYPDLCS	A	I	[87]
		Pl170	GIGGSCCVIRSCAIKFSTLCG	A	I	[87]
		Pl172	TCYGVCLEDKKPEEHCWEEVTKTVRGEPGDVQFC	A		[87]
		α/κ-PlXIVA	FPRPRICNLACRAGIGHKYPFCHCRX	J	XIV	[87]
		Pl058	FPRPRICNLACRAGIGYKYPFCHCR	J		[87]
		Pl022	SFDCCPQYDYCCW	T		[87]
		κ-Y-Pl1	ARFLHPFQYYTLYRYLTRFLHRYPIYYIRY	Conopeptide Y		[87]
		Pl069	FQSWPLTNPDLKAAFVKGSAQRVAHGYG	NSf-1		[87]
		Pl074	AFKQYNWQRMPYGT	NSf-1		[87]

**Table 15 toxins-16-00094-t015:** αD-conotoxins identified from predatory and defense venoms of worm-hunting cone snails. Presented here are conotoxins found exclusively in the predation-evoked or the defense-evoked venoms. Cysteine residues are highlighted in **red**. The complete table can be found in the Appendix A.

Clades	*Conus* Species	Conotoxins	Mature Sequence	Gene Superfamily	Cysteine Framework	References
*Stephanoconus*	*imperialis*	Im28.1	LHCHEISDLTPWILCSPEPLCGGKGCCAQEVCDCSGPVCTCPPCL	D	XXVIII	[9]
*Rhizoconus*	*vexillum*	α-VxXXA	DVQDCQVSTPGSKWGRCCLNRVCGPMCCPASHCYCVYHRGRGHGCSC	D	XX	[86]
		α-VxXXB	DDESECIINTRDSPWGRCCRTRMCGSMCCPRNGCTCVYHWRRGHGCSCPG	D	XX	[86]
		α-VxXXC	DLRQCTRNAPGSTWGRCCLNPMCGNFCCPRSGCTCAYNWRRGIYCSC	D	XX	[86]
	*capitaneus*	Cp20.3	EVQECQVDTPGSSWGKCCMTRMCGTMCCSRSVCTCVYHWRRGHGCSCPG	D	XX	[86]
		Cp20.5	DNEAECQIDTPGSSWGKCCMTRMCGTMCCSRSVCTCVYHWRRGHGCSCPG	D	XX	[86]

## Data Availability

All data used in this study were retrieved from publicly available databases.

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
