# Peer review of "Predatory and Defensive Strategies in Cone Snails"

_toxins, 2024, doi:10.3390/toxins16020094_

Round 1

Reviewer 1 Report

Comments and Suggestions for Authors

General comment. 

Overall, a comprehensive summary of the literature of cone snail venom composition. However, the review is dotted with minor factual, grammatical and structural errors which can be easily rectify.

Abstract.

Line 18: Replace “Rhizoconus and Vituliconus” with “Rhizoconus and Vituliconus

Introduction.

Line 42: Replace “receptors in the nervous system, such as ion channels and other transmembrane proteins” with “membrane receptors, ion channels and other transmembrane proteins of the nervous and non-nervous systems”

Line 47: Replace “mature peptide” with “mature peptide (Figure 1)”

Figure 1 legend: Replace “ω-SVIA (Conus striatus), ω-CVID (Conus catus), ω-MVIIA (Conus magus), δ-PVIA (Conus purpurascens), δ-TxVIA (Conus textile)” with “ω-SVIA (C. striatus), ω-CVID (C. catus), ω-MVIIA (C. magus), δ-PVIA (C. purpurascens), δ-TxVIA (C. textile)”

Line 60: Replace “GPCRs” with “G-protein could receptors”

Line 62: Replace “Figure 1” with “Figure 2”

Line 66: Replace “Conus” with C.

Line 67: Replace “nAChRs” with “nicotinic acetylcholine receptors (nAChRs)”

Line 68: Delete “to help”

Figure 2: Replace “NE” with “Norepinephrine (NE)”

Line 91: Replace “cones” with “cone snails”

Line 93: Replace “cone” with “cone snail”

Line 95: Replace “cone” with “cone snail”

Line 109: Replace “cones” with “cone snails”

Line 110: Replace “cone” with “cone snail”

Line 111: Replace “cone” with “cone snail”

Line 112: Replace “cones” with “cone snails”

Figure 4 legend: Delete “look”

Line 127: Replace “implies” with “involves”

Line 131: Replace “MS” with “mass spectrometry (MS)”

Line 137: Replace “C. imperialis and C. regius both” with “Both C. imperialis and C. regius

Line 170: Replace “was ingeniously devised a procedure: with “a procedure was devised ingeniously,”

Line 209: Replace “cone” with “cone snail”

Line 257: Replace “cones” with “cone snails”

Line 257-258: Replace “Conus” with “C.”

Line 265: Replace “cones” with “cone snails”

Line 277: Table 1 should come first

Line 285: Replace “Pionoconus” with “Pionoconus

Line 290: Replace “Mayotte” with “Mayotte (France)”

Line 317: Perhaps include a short explanation for the m/n classification?

Line 319: Replace “neuronal” with “non-muscle”

Line 400: Replace “AMPA” with “AMPA (α-amino-3-hydroxy-5-methyl-4-isoxazole propionic acid)”

Line 444: Table 7 appears before line 443. Table 7 should come after Line 443-451 paragraph.

Line 456: Replace “cones” with “cone snails”

Line 481: Replace “cone” with “cone snail”

Line 484: Replace “cones” with “cone snails”

Figure 12 legend: Replace “(Conus marmoreus (Conus), Conus ammiralis, Conus victoriae (Cylinder). No defense venoms were described from Conus textile” with ““(C. marmoreus (Conus), C. ammiralis, C. victoriae (Cylinder). No defense venoms were described from C. textile

Line 587: Replace “less” with “least”

Line 589: Replace “etc)” with “etc),”

Line 590: Replace “And this lack of investigation consequentially” with “Consequentially, this lack of investigation”

Line 591: Replace “cones” with “cone snails”

Line 621: Replace “α7” with “the α7”

Line 647: Replace “C. elegans” with “Caenorhabditis elegans

Line 651: Replace “important, driving forces (predation and defense)” with “important driving forces, predation and defense”

Line 716: Replace “AI” with “AI (artificial intelligence)”

Comments on the Quality of English Language

Minor grammatical and structural errors which can be easily rectify.

Author Response

See responses in the attached pdf

Reviewer 2 Report

Comments and Suggestions for Authors

Thank you for this very interesting contribution - I enjoyed reading it.

I have come at this review not from a background of being a cone snail or conotoxin specialist, but from the perspective of a marine biologist who has maintained a deep-rooted interest in marine venoms that has lasted for 25 years or so. I also teach this topic at undergraduate level and I am therefore very interested to see any new and accessible literature on conotoxins, particularly if I feel they could make an educational as well as a research knowledge contribution.

I will make some general observations below but I would ask the authors to carefully check the attached annotated copy of the manuscript - this includes comments, points of clarifications and suggestions for improving some aspects of the text. I will not repeat these here.

Before I get on to the more interesting points I would like to draw the author's attention to some issues with the structure of their manuscript, specifically relating to the Figures and Tables.

For the Figures, it was notable that the first instance that Figure 2 was cited in the text was on line 315 - by this point the manuscript was already up to citing Figure 8. So, either cite Figure 2 much earlier in the manuscript or you will need to rearrange your figure order. Figure 15 wasn't cited in the text at all. There was also variation in your use of bold for the in-text Figure citation and there was also variation in your use of upper or lower case letters to identify figure panels. Please revise the manuscript carefully to ensure the approach is consistent throughout.

There were a lot of tables in the manuscript (15 in all), and there were similar issues with the order in which you referred to the tables in the text. The order as it stands is 2, 1, 3, 4, 5, 6, 7, 10, 12, 13, 8, 9, 11, 14, 15. Really, Tables (as per figures) should be cited in the order that they are presented in the manuscript. Please correct this.

Okay, on to more constructive discussion. I was very excited to receive the invitation to review this manuscript as I hold the view that the study of cone snail ecology has been all but lost among the tidal wave of studies focused on characterising venom composition. Although not explicitly stated in the manuscript, reading between the lines it would at least indicate that the authors are acknowledging this mismatch in research effort. So, the prospect of a review article that could redress this imbalance has a lot of appeal. 

However, I'm not sure that the manuscript entirely delivered as I had hoped (perhaps that's down to unrealistic expectations on my part, perhaps not). The authors very much took a track whereby they tried to derive ecological meaning from the compositional change of venom from cone snails across the classical feeding guilds of fish, snail, and worm eaters. This approach was underpinned by a very interesting and very relevant consideration of offensive versus defensive venoms - as this is a relatively new direction in cone snail venom research it is to be expected that many data and knowledge gaps exist. However, as an 'old fashioned' marine biologist brought up on the basics of marine ecology, I didn't necessarily find the answers I was looking for. It could be that those answers don't exist, but I am going to pose a question here for the authors to consider and, if they are aware of literature that either supports or refutes my question, I would ask them to include it in their manuscript.

One of the basic lessons that was taught to me as a student studying cone snail venom 25 years ago was that the worm eaters were considered the most 'primitive' in terms of their venom potency (my Professor's words, not mine), followed by the snail eaters and then the fish eaters. My Professor argued that this disparity in venom potency arose primarily due to the potential for a given prey item to flee beyond the sphere of influence of the cone snail, in other words a stung fish could, in a very short period of time, swim out of the snails reach and could be consumed by another predator (granted, this view does seemingly discount the tethering of the radula to the cone snail). By extrapolation, the slower moving snails and worms would need less powerful venom. Inevitably this is a crude over simplification, but I do wonder whether anyone has systematically tried to deconstruct this theory? Given that this manuscript is trying to take a quasi ecological view of venoms in cone snails I would put it to the authors that it might be worthwhile addressing my point - I would certainly be very interested in their perspective and I'm sure many a student of marine venom ecology would also be interested.  

The authors also sew doubts about the validity of venom cabals - I don't have sufficient subject-level expertise to critique this but I would perhaps ask the authors to retain an open mind. By all means question the evidence base (as they have), but there are ways to formulate your scepticism that doesn't necessary belittle those who have and continue to publish work on cabals - just something for the authors to think about.

Anyway, an enjoyable and educational manuscript - please respond to the points given on the annotated pdf. If accepted for publication (following revision) I look forward to including the paper in the reading list for my venoms course.

Comments on the Quality of English Language

Very minor revisions needed - these have been marked up on the annotated pdf

Author Response

See responses in the attached pdf.

Reviewer 3 Report

Comments and Suggestions for Authors

Authors reported a comprehensive review on the predatory and defensive strategies in cone snails. Apart from morphology and ecological interactions, authors described the pharmacology and biological activities of venoms and toxins in the context of the cone snails predatory and defensive strategies. This is a very well written articles, easy to follow, and add insights in the related field. Questions unresolved to date were discussed, and suggestions were given for directions of future research. There are only some minor comments and suggestions, as given below, for author(s) consideration to improve the article. 

Line 38-39: it was stated that the peptides are characterized as conopeptides or conotoxins. The subsequent sentence Line 39-40 stated that conotoxins and conopeptides are clearly different (one being cysteine-rich, one with 1 or none). If cone snail venom contains both groups of peptides, then the first statement should use "and" instead of "or". For the subsequent statement, are there any specific cysteine-poor cono-peptides to be given as examples to contrast conotoxins? 

Line 43: "signal molecule" - can this be made more specific? Do you mean "ligand"? 

Figure 1: Suggest to include the disulfide bond pairing on the aligned sequences. 

Secondly, can authors try to code the amino acid residues with color shade to indicate their conservation and divergence? 

Line 62: etc. (one full stop)

Figure 2 caption may benefit from further elaboration. Brief description should be given to annotate the classifying lines, alphabets and roman numbers.

Figure 3 (and other): Can authors provide a color version for the drawing?

Lines 93, 109 and onward: It seems that authors replaced "cone snails" with "cones" throughout the text. For a scientific writeup as such, it may be better to standardize the terminology and follows that of the standard term used in the field. 

Line 161-164: Were these "hormone-like peptides" identified or characterized, and with regard to the "insulin-like toxins/peptides" supposedly used by some cone snails to create the hypoactive state? For example in this paper:  DOI: 10.1002/prot.26265. Accordingly, is "hypoglycemia-inducing" one of the evolved biochemical strategy used by certain cone snail species? 

Line 192 onward - Under this section (defensive strategies), suggest to include envenoming in the context of human having been "stung". I am not sure if cases have been reported, but anecdotally there were a number, for instance, divers who harvested and kept the snails close to the body.

Discussion: 

Line 657 - "...pooling large batches of injected venom could also "falsely" increase venom complexity..." -- in this context, can authors discuss the challenge faced in collecting the cone snail venom to an amount sufficient for research, and what advice/suggestion might authors give to juniors entering this field.

Lastly, as authors mentioned the advent of mass spec technologies and Big Data surely benefit the research in this field tremendously in the future. What is the status of "venomics" or genomics in relation to cone snake venom research? This may be added in the current work as a comprehensive review. 

Author Response

See responses in the attached pdf.
